



# Physical and chemical properties of deposited airborne particulates over the Arabian Red Sea coastal plain

Johann Engelbrecht[1,2], Georgiy Stenchikov[1], P. Jish Prakash[1], Anatolii Anisimov[1] and Illia Shevchenko[1]

[1]King Abdullah University of Science and Technology (KAUST), Physical Science and Engineering Division (PSE), Thuwal, 23955-6900, Saudi Arabia.

[2]Desert Research Institute (DRI), Reno, Nevada 89512-1095, U.S.A.

*Correspondence to:* Johann P. Engelbrecht (johann@dri.edu)





**Abstract**

Mineral dust is the most abundant aerosol, having a profound impact on the Global energy budget. This research continues our previous studies performed on surface soils in the Arabian Peninsula and aims at analyzing mineralogical, physical and chemical composition of dust deposits from the atmosphere. For this purpose, aerosols deposited from the atmosphere are collected during 2015 at six sites on the campus of the King Abdullah University of Science and Technology (KAUST) situated on the Red Sea coastal plain of Saudi Arabia and subjected to the same chemical and mineralogical analysis we conducted on soil samples. Frisbee deposition samplers with foam inserts were used to collect dust and other deposits, for the period December 2014 to December 2015. The average deposition rate measured at KAUST for this period was 14 g m$^{-2}$ per month, with lowest values in winter and increased deposition rates in August to October. XRD analysis of a subset of samples confirms variable amounts of quartz, feldspars, micas, and halite, with lesser amounts of gypsum, calcite, dolomite, hematite, and amphibole. Freeze-dried samples were re-suspended onto the Teflon® filters for elemental analysis by XRF, while splits from each sample were analyzed for water soluble cations and anions by Ion Chromatography. It is not possible to explicitly relate the origin of deposition samples to the Arabian Red Sea coastal soils, from the mineralogical and chemical results alone. It is proposed that the dust deposits along the Red Sea coast are a mixture of dust emissions from local soils, and soils imported from distal dust sources. These aerosol data are the first of their kind from the Red Sea region. They will help assess their potential nutrient input into the Red Sea, as well the impact on human health, industry, and solar panel efficiency. These data will also support dust modeling in this important dust belt source area, by better quantifying dust mass balance and optical properties of airborne dust particles.

**Keywords**

Dust mineralogy, chemistry, particle size distribution

Inverted Frisbee deposition samplers

Freeze-dry dust retrieval

Dust deposition, AOD and visibility




## 1. Introduction

Dust emission and deposition modeling and measurements are required for the assessment of the dust mass budget. Both emission and deposition are under constrained in atmospheric dust models, leading to large uncertainties (Bergametti and Forêt, 2014). To improve simulations, these authors

are, amongst others, suggesting the establishment of dust deposition networks in the vicinity of and away from dust source regions, operating throughout the year. In this paper we are presenting results from a network of dust deposition samplers located on the campus of the King Abdullah University of Science and Technology (KAUST) along the Red Sea coast of Saudi Arabia. This is an important dust source region, the effect of which extends thousands of kilometers downwind.

To better characterize optical, microphysical, and health effects of dust aerosols we have conducted detailed chemical and mineralogical analysis of deposition samples collected from atmosphere.

Mineral dust is the most abundant atmospheric aerosol, primarily from suspended soils in arid and semi-arid regions on Earth (Buseck et al., 2000; Washington and Todd, 2005; Goudie, 2006; Muhs

et al., 2014), including deserts of the Arabian Peninsula (Edgell, 2006). Dust aerosols profoundly affect climate, biogeochemical cycles in the ocean and over land, air-quality, atmospheric chemistry, cloud formation, visibility, and human activities (Bennett et al., 2006; Bennion et al., 2007; De Longueville et al., 2010; Fryrear, 1981; Hagen and Woodruff, 1973; Haywood and Boucher, 2000; Hsu et al., 2004; Jickells et al., 2005; Kumar et al., 2014; Mahowald, 2009;

Menéndez et al., 2017; Nihlen and Lund, 1995; Prospero et al., 2002; Sokolik and Toon, 1999; Twomey et al., 1984; Wang et al., 2010; Huang et al., 2006). The Arabian Peninsula is one of Earth's major sources of atmospheric dust, contributing as much as 11.8% (22–500 Mt a$^{-1}$) of the total (1,877–4000 Mt a$^{-1}$) global dust emissions (Tanaka and Chiba, 2006). The Red Sea, being enveloped by the Arabian and African deserts is strongly impacted by windborne mineral dust.

Along with profound influence on the surface energy budget over land and the Red Sea (Kalenderski et al., 2013; Osipov et al., 2015; Brindley et al., 2015), dust is an important source of nutrients, more so for the oligotrophic northern Red Sea waters (Acosta et al., 2013). From preliminary assessments it is estimated that 5 to 6 major dust storms per year impact the Red Sea region, depositing about 6 Mt of mineral dust into the Red Sea (Prakash et al., 2015). Simulations

and satellite observations suggest that the coastal dust contribution to the total deposition flux into



the Red Sea could be significant, even during fair weather conditions (Jiang et al., 2009; Anisimov et al., 2017). Therefore, the correct representation of the regional dust balance over the Red Sea coastal plain is especially important. Here we specifically focus on the dust deposition in this area, which helps to constrain the dust mass balance, as well as the dust mineralogy and chemical

composition. Dust sources impacting on the Arabian Red Sea coastal region were shown to vary by season, coming from local haboobs and low level jets, delivered from the Tokar delta of Sudan in summer (Kalenderski and Stenchikov, 2016), and transported from the west coast of the Arabian Peninsula (Kalenderski et al., 2013).

The mineralogy and chemical composition of dust generated from the Red Sea coastal region

remains uncertain. The Red Sea coastal plain is a narrow highly heterogeneous piedmont area, and existing soil databases do not have enough spatial resolution to represent it adequately (Nickovic et al., 2012).

Minerals previously identified in continental soils from Middle East dust generating regions include quartz, feldspars, calcite, dolomite, micas, chlorite, kaolinite, illite, smectite, palygorskite,

mixed-layer clays, vermiculite, iron oxides, gypsum, hornblende and halite (Engelbrecht et al., 2009a; Engelbrecht et al., 2016; Goudie, 2006; Prakash et al., 2016; Pye, 1987; Scheuvens and Kandler, 2014). It could be expected that similar mineral assemblages would occur in variable proportions in the dust deposition samples collected in the region.

With the exception of the area around Jazan in the south, which is impacted by the Indian Ocean

monsoon, the Red Sea coastal region has a desert climate characterized by extreme heat, reaching 43° C during the summer days, with a drop in night-time temperatures on average more than 10° C. Although the extreme temperatures are moderated by the proximity of the Red Sea, summer humidity is often 85 % or higher during periods of the northwesterly *Shamal* winds. Rainfall diminishes from an annual average of 133 mm at Jazan in the south to 56 mm at Jeddah, and 24

25    mm at Tabuk in the north. Vegetation is sparse, being restricted to semi-desert shrubs, and acacia trees along the ephemeral rivers (wadis), providing forage for small herds of goats, sheep and dromedary camels.

During infrequent but severe rainstorms, run-off from the escarpment along wadis produce flash floods in lowland areas. With such events, fine silt and clay deposits are formed on the coastal





plain, which are transformed into dust sources during dry and windy periods of the year. The resultant dust is transported and deposited along the coastal plain itself and adjacent Red Sea, by prevailing northwesterly to southwesterly winds, with moderate breezes (wind speed >5.5 m s$^{-1}$) from the north (http://www.windfinder.com/weather-maps/report/saudiarabia#6/22.999/34.980).

The importance of dust mineralogy was long been recognized, but only recently the explicit transport of different mineralogical species is implemented in climate models (Perlwitz et al., 2015a, b).

Another important implication of dust emission/deposition processes is associated with the harvesting of the solar renewable energy in the desert areas. However, dust deposits on solar panels

are known to have a severe detrimental effect on the efficiency of photovoltaic systems (Goossens and Van Kerschaever, 1999; Hamou et al., 2014; Mejia et al., 2014; Rao et al., 2014; Sulaiman et al., 2014; Ilse et al., 2016), varying with mineral composition and atmospheric conditions (Supplement).

## 2.  Objectives

This study is meant to complement the recently published papers by our research group that characterize the effect of dust storms (Prakash et al., 2015; Kalenderski et al., 2013), evaluate radiative effect of dust (Osipov et al., 2015), analyze soils from the Red Sea coastal plain (Prakash et al., 2016) and dust emissions in the same region (Anisimov et al., 2017). It aims to provide mineralogical, physical and chemical compositions of deposition samples collected largely during

2015 at six sites on the campus of KAUST, located approximately 80 km north of Jeddah, along the central part of the Red Sea coastal plain of Saudi Arabia, (Fig. 1).  The coastal plains of the Arabian Peninsula along the Red Sea and Persian Gulf are among the most populated areas in this region, hosting several major industrial and residential centers. Airborne dust profoundly affects human activities, marine and land ecosystems, climate, air-quality, and human health. Satellite

observations suggest that the narrow Red Sea coastal plain to be an important dust source province, augmented by the fine-scale sediment accumulations, scattered vegetation, and variable terrain. Airborne dust inevitably carries the mineralogical and chemical signature of a parent soil. The purpose of a previous study on 13 soil samples from the Arabian Red Sea coastal area (Prakash et al., 2016) was to better characterize their mineralogical, chemical and physical properties, which



in turn improve assessment of dust being deposited in the Red Sea and on land, affecting environmental systems and urban centers. Prakash et al. (2016) found that the Red Sea coastal soils contain major components of quartz and feldspar, as well as lesser but variable amounts of amphibole, pyroxene, carbonate, clays, and micas, with traces of gypsum, halite, chlorite, epidote

and oxides. The mineral assemblages in the soil samples was ascribed to the variety of igneous and metamorphic provenance rocks of the Arabian Shield forming the escarpment to the east of the coastal plain.

The specific objective of the present study is to examine mineralogical and chemical information from deposition samples collected on the KAUST campus. This will help to better quantify the

ecological impacts, health effects, damage to property, and optical effects of dust blown across this area (Engelbrecht et al., 2009b, a; Weese and Abraham, 2009). The obtained chemical and mineralogical information allows to assess the likelihood that deposited dust at least in part originates from local sources and in part from remotely transported dust. Knowledge of the mineralogy of the dust deposits will provide information on refractive indices, which can be used

to calibrate dust radiative transfer models, and help assessing the impact of dust events on the Red Sea and adjacent coastal plain. This research complements our dust studies performed in the Arabian Peninsula (Engelbrecht et al., 2009b; Kalenderski et al., 2013; Prakash et al., 2015; Prakash et al., 2016) as well as globally (Engelbrecht et al., 2016).

### 3.    Sampling and analysis

Anisimov et al. (2017) estimated that the eastern Red Sea coastal plain emits about 5–6 Mt of dust annually. Due to its close proximity, a significant portion of this dust is likely to be deposited into the Red Sea, which could be comparable in amount to the estimated annual deposition rate from remote sources during major dust events  (Prakash et al., 2015). Therefore, we expect that the total dust deposition into the Red Sea is of the order of 10 Mt a$^{-1}$, but this figure still needs to be

confirmed.

In the past few decades wind tunnel and field tests to compare their efficiencies, had been performed on different designs of deposition samplers and sand traps, including marble dust collectors (MDCO), inverted Frisbees, and glass surfaces (Goossens and Rajot, 2008; Sow et al., 2006; Goossens et al., 2000; Goossens and Offer, 2000). Most of the experiments performed in

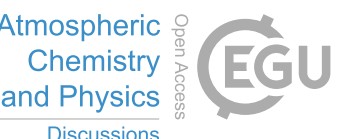

wind tunnels failed to completely mimic the field conditions, which resulted in an underestimation of the dust deposition, more so for the <10 μm size fraction (Sow et al., 2006). Based on the field evaluations by Vallack (1995) and suggestions by Vallack and Shillito (1998) the decision was taken to deploy inverted Frisbee samplers with foam inserts.

At each sampling site the particulate deposits were collected into a 227 mm diameter inverted Frisbee dust deposit sampler, each with a polyester foam insert and bird strike preventers (Hall et al., 1993; Vallack and Chadwick, 1992, 1993; Vallack and Shillito, 1998) (Figure 2). The purpose of the foam insert is to enhance the particulate collection capacity of the dust gauge (Vallack and Shillito, 1998) by better collecting and retaining wet (from fog, dew, rain) and dry, fine and coarse
particles being deposited into the inverted Frisbee dish, under stable meteorological conditions, during severe dust events, northwesterly *Shamal* winds, and by daily coastal winds,

For the period December 2014 to March 2015, four Frisbee samplers were located at the New Environmental Oasis (NEO) site, about 50 m apart. The gravimetric information from the four samplers were similar, with small variations amongst them ascribed to the impact from local
construction activities. Due to the similarity of these gravimetric results, and to obtain a better representation of dust deposition onto the KAUST campus, two of the samplers (DT1 and DT2) were moved in March, the first (DT1) to a residential area and the other (DT2) to the quay adjacent to the Coastal & Marine Resources Core Lab (CMOR) (Table 1). (Site meta-data provided in the Supplement).

The deposition samples were collected for intervals of a calendar month, starting in December 2014 and ending December 2015. At the end of each month, the samples are retrieved by flushing the dust deposit from the foam insert and collection dish into the downpipe and plastic bottle, with distilled water. Both the insoluble particles and dissolved salts in the water suspension are retrieved in the laboratory by a freeze-drying (sublimation) procedure.

A total of 52 deposition samples were collected at the six sampling sites on the KAUST campus (Fig. 2) over a period of 13 months, largely in 2015. Representative subsets of these were selected for X-ray diffraction (XRD), (27 samples) and chemical analysis (29 samples).





Freeze-dried sample splits were re-suspended in the laboratory onto Teflon® filters, for elemental analysis by X-ray Fluorescence (XRF) spectrometry by using a miniaturized version of a dust entrainment facility (Engelbrecht et al., 2016) http://www.dri.edu/atmospheric-sciences/atms-laboratories/4185-dust-entrainment-and-characterization-facility ). With this modified system the dust sample is drawn into a vertically mounted tubular dilution chamber, and the re-suspended dust collected onto a 47 mm diameter Teflon® filter, for chemical analysis.

The samples re-suspended onto the Teflon® filters were chemically analyzed for elemental content by XRF, including for Si, Ti, Al, Fe, Mn, Ca, K, P, V, Cr, Ni, Cu, Zn, Rb, Sr, Y, Zr, and Pb (US EPA, 1999). Splits of about 2 mg from each frieze-dried sample was analyzed for water soluble cations of sodium ($Na^+$), potassium ($K^+$), calcium ($Ca^{2+}$) and magnesium ($Mg^{2+}$), and anions of sulfate ($SO_4^{2-}$), chloride ($Cl^-$), phosphate ($PO_4^{3-}$) and nitrate ($NO_3^-$), by Ion Chromatography (IC) (Chow and Watson, 1999).

A subset of 27 samples from the total of 52 samples, representing all months of the year, was selected for X-ray diffraction (XRD) analysis. XRD is a non-destructive technique particularly suited to identifying and characterizing minerals such as quartz, feldspars, calcite, dolomite, clay minerals and iron oxides, in fine soils and dusts. Dust reactivity in seawater as well as optical properties depend on its mineralogy, e.g. carbonates and sulfates are generally more soluble in water than silicates such as feldspars, amphiboles, pyroxenes or quartz. A Bruker D8® X-ray powder diffraction system was used to analyze the mineral content of the dust deposition samples. The diffractometer was operated at 40 kV and 40 mA, with Cu Kα radiation, scanning over a range of 4-50° 2θ. The Bruker Topas® software and relative intensity ratios (RIRs) were applied for semi-quantitative XRD analyses of the dust deposition samples (Rietveld, 1969; Chung, 1974; Esteve et al., 1997; Caquineau et al., 1997; Sturges et al., 1989).

## 4. Results

### 4.1 Meteorology

Northwesterly *Shamal* winds prevailed during all twelve months of 2015 (Fig. 3). Four to five severe dust storms lasting three to five days each, contributed to hot humid conditions during the summer months. Also, weaker north-easterly winds were experienced in October and November of that year.




The first four months in the second half of 2015 experienced the highest ambient temperatures (Table 2), with an average temperature of 35º C for August followed by 34º C for September, bracketed by 33º C for both July and October. The highest single temperature was 43º C, recorded in October, with the coolest temperature of 17º C in January of that year. The range of temperatures

was the greatest through fall, winter and spring, with large diurnal temperature fluctuations during these seasons. The humidity at KAUST is consistently high (Table 2), with averages varying from 57 % for December and 61% for January, to as high as 82 % for August and 80% for September. Dewpoints were calculated for each set of hourly measurements, applying the August-Roche-Magnus approximation (Alduchov and Eskridge, 1996; August, 1828; Magnus, 1844). The highest

dewpoint temperatures were calculated in August (31º C) and September (30º C) while the month with the greatest frequency of humidity measurements (96) in excess of 90 % was also recorded in August (Table 2, Fig. 4). The lowest monthly frequency (4) for humidity exceeding 90 % was December.

## 4.2 Gravimetric analysis

With a few exceptions, the monthly gravimetric measurements from the four samplers (DT1 – DT4) are comparable (Fig. 5), fluctuating similarly by month and season. The deposition rates were at their lowest for December 2014 (avg. 4 g m$^{-2}$), increasing steadily for four months to a peak value for March, 2015 (avg. 20 g m$^{-2}$) before decreasing over the subsequent four months to a low for July (avg. 5 g m$^{-2}$). The deposition rates increased sharply for August (28 g m$^{-2}$),

September (23 g m$^{-2}$) and October (28 g m$^{-2}$), before diminishing in November (14 g m$^{-2}$) and December (11 g m$^{-2}$). The NEO terrain is close to several building construction sites, about 400 m to the east and southeast of the installed deposition samplers, which periodically created substantial amounts of local airborne dust. This, together with the windy conditions are held responsible for elevated dust concentrations measured at the two NEO sites (DT3, DT4). Wind-blown sea spray

during stormy conditions was responsible for elevated deposition levels of sea salt at the CMOR (DT2) quay-side site, for the months of September and October 2015.

Bearing in mind that the dust deposition samplers, sampling procedures, as well as conditions and sampling periods were different to those of this study, some comparisons to similar studies in desert regions are listed in Table 3. The deposition rates from this study, both on average (14 g m$^{-2}$ month$^{-1}$)

and in range (4-28 g m$^{-2}$ month$^{-1}$), were found to be similar to those previously recorded





by Offer and Goossens (2001) in the Negev Desert, Israel (average 17 g m$^{-2}$ month$^{-1}$, range 10-25 g m$^{-2}$ month$^{-1}$), and West Niger (Goossens and Rajot, 2008) (average 13 g m$^{-2}$ month$^{-1}$, range 6-21 g m$^{-2}$ month$^{-1}$). A campaign in the Saudi Arabian capital of Riyadh (Modaihsh, 1997; Modaihsha and Mahjoub, 2013) during the dusty months of January to March showed average

5 monthly deposition rates of 42 g m$^{-2}$ , and a range of 20-140 g m$^{-2}$. The dust deposition measured in Kuwait on the other hand, varied substantially by sampling district, being impacted to a variable extent by the disturbed soils from the Mesopotamian lowlands and the northwesterly *Shamal* winds. For these reasons the dust loadings varied during the campaigns in the N-E Bay area of Kuwait during the 2002/3 period (Al-Awadhi, 2005) (average 28 g m$^{-2}$ month$^{-1}$, range 3-58 g m$^{-2}$

10 month$^{-1}$) to samples collected in Kuwait City during the 2011/12 period (Al-Awadhi and AlShuaibi, 2013) (average 53 g m$^{-2}$ month$^{-1}$, range 2-320 g m$^{-2}$ month$^{-1}$) and samples collected during the 1979/80 campaign in the N-W Gulf area of Kuwait (Khalaf and Al-Hashash, 1983) (average 191 g m$^{-2}$ month$^{-1}$, range 10-1003 g m$^{-2}$ month$^{-1}$).

### 4.3  AERONET and visibility measurements

An CIMEL Robotic Sun Photometer is installed on the rooftop of the CMOR building on the campus of the KAUST and operated by our group since 2012, as a part of the NASA- AERONET, providing aerosol optical depth (AOD) and aerosol retrieved characteristics (https://aeronet.gsfc.nasa.gov/). Figure 6 compares the monthly averaged AOD at 500 nm with the dust deposition rate for 2015. In a general sense the AOD and the deposition rates show

comparable trends, both with maxima in spring and larger maxima in fall. However, the AOD reaches a first maximum in April, being one month later than that of the deposition rate. Also, the larger second AOD maximum occurred in August while the maximum deposition rate is broadly distributed over a three-month period, from August to October. The photometer measures light attenuation by all aerosols along a column in the atmosphere, while deposition rate depends on

dust at ground level only, the latter generally containing a relatively coarser dust fraction. The low-level dust particles are predominantly from local dust sources while the higher altitude dust could be transported from distal sources and chemically transformed, i.e., aged. As was pointed out by Yu et al. (2013) the differences between the deposition and AOD time series can in part be attributed to modifications of the natural dust aerosol by anthropogenic activities, including

petrochemical and other large industries along the Red Sea coast, as well as by entrainment of





construction and road dust. The linear correlation coefficient between the monthly deposition rates and the monthly averaged AOD of 0.40 suggests a causal interrelationship between these two quantities.

Furthermore, a comparison between the deposition samples and the visibility is made with measurements taken in 2015 at the Jeddah airport meteorological station, approximately 70 km to the south of KAUST. Visibility is expressed as the frequency of dust events with reported weather codes 06-09, or 30–35, grouped as dusty or non-dusty days, for each month (Notaro et al., 2013; Anisimov et al., 2017), expressed as percentages. The bimodal monthly distributions seen with the deposition rates and AERONET monitoring are also mirrored by the visibility measurements collected at Jeddah (Figure 7). The linear correlation coefficient between the monthly deposition rates and monthly averaged visibility measurements is 0.48, clearly suggesting a causal relationship between the two variables. It has been suggested that visibility and visibility frequency be used as a metric of dust emission flux and near surface dust concentrations (Cowie et al., 2014; Anisimov et al., 2017; Yu et al., 2013; Norris et al., 2014; Shao and Dong, 2006; Notaro et al., 2013).

### 4.4   Mineral analysis by XRD

XRD analysis of the 27 samples (Fig. 8) show variable amounts of quartz (6–38 %, avg. 22 %) and feldspars (plagioclase, K-feldspar) (5-34 %, avg. 20 %), clays (10-18 %, avg. 13 %), micas (6-31 %, avg. 13 %), halite (1-53 %, avg. 7 %) with lesser amounts of gypsum (1-8 %, avg. 4 %), calcite (0-8 %, avg. 2 %), dolomite (0-7 %, avg. 3 %), hematite (0-8 %, avg. 3 %), and amphibole (and pyroxene) (0-4%, avg. 1 %).

From the XRD, four broad mineral assemblages can be distinguished, the first and major assemblage is comprised of feldspars, clays and micas as well as hematite and gypsum, the second group is of quartz, the third of halite, and the fourth of calcite.

There is an increase in the halite concentrations at sites DT1-DT3, from about 2 % (DT1) in December 2014 to about 53 % (DT2) in July 2015 (Fig. 8). From August onwards there is an abrupt decrease in halite content to less than 5 %, except for samples collected at the DT2 (CMOR, quay-side) site alongside the ocean. There was a simultaneous increase in the proportion of quartz to a maximum of 38 % in April (DT3), and decreasing to less than 25 % at all sites after July,





2015. The silicate mineral group decreased systematically from about 72 % (DT1) in December 2014 to about 25 % (DT2) in July. Except for two samples from the DT3 site collected in September and October 2015, the dominant minerals after July, 2015 included the silicate assemblage, with concentrations of up to 80 %. The variation in the proportions of the four mineral

assemblages, especially the halite, is ascribed to seasonal fluctuations in wind, humidity and precipitation, as well as the proximity of the sea to the samplers.

## 4.5   Particle size distribution

Dust deposition rates are dependent on both the meteorological conditions and dust properties such as particle size distribution. The size distributions of surface soils from potential dust source

regions along the Red Sea coastal plain were previously documented (Prakash et al., 2016). The size distribution as measured on one deposition sample (DT1) (Fig. 9a) by SEM differs marginally from the size distributions of the soils from the coastal dusts (Fig. 9b). The deposition sample with a geometric mean particle diameter of 2.9 μm is slightly coarser than the average <38 μm sieved soils with a geometric mean diameter of 2.5 μm. The deposition sample also shows a broader

distribution towards the larger particle sizes. This marginal difference can be ascribed to a predominant deposition of a coarser particles (Prakash et al., 2016). In addition the Frisbee sampler is biased towards the sampling of the coarser particles, as previously documented (Bergametti and Forêt, 2014; Goossens, 2005).

The deposition samplers collect the total suspended particulates (TSP) but the bin aerosol models

usually calculate only $PM_{10}$. So they simulate deposition rates that should be lower than that we observed. However, the size distribution of deposited particles shown in Figure 9 could be used to distinguish the contribution of $PM_{10}$ in deposited mass and reconcile models with observations.

## 4.6   Chemistry (XRF and IC)

As expected, the chemically analyzed deposition samples contain major amounts of $SiO_2$

(Appendix A, Fig. 10a, b), varying between 12–53 % (avg. 31 %) in the sample subset, occurring as quartz, and together with $Al_2O_3$, (avg. 4 %) and CaO, (avg. 2.3 %) in plagioclase, and $K_2O$ (avg. 0.6 %) in potassium feldspars. $SiO_2$ together with $Al_2O_3$, $Fe_2O_3$, $TiO_2$, MnO, MgO, and some $K_2O$ are also contained in the clays, micas and amphiboles, previously identified in these samples by optical microscopy and XRD. Lesser amounts of CaO are contained in gypsum and calcite, and





together with MgO, in dolomite. The iron expressed here as $Fe_2O_3$ can be contained in hematite ($Fe_2O_3$), goethite FeO(OH) or in clay minerals such as illite, each with different solubility. It has been suggested that a large fraction of iron in soils and dusts is contained as amorphous colloidal coatings on quartz and feldspars (Engelbrecht et al., 2016).

5    The water-soluble cations (Appendix A, Fig. 11a, b) account for 1-19 % and the anions for 1-30 % of the total mass, respectively. These account for variable amounts of halite (1-32 %), and gypsum (1-9 %), with lesser amounts of other chlorides and carbonates. Of importance as dust borne nutrients likely to be deposited in the Red Sea, are the low concentrations of both water soluble $NO_3^-$ (avg. 0.8 %), and water soluble $PO_4^{3-}$ (avg. 0.2 %) compared to the total $P_2O_5$ (avg.
10    0.3 %) in the dust deposits. The phosphorus is contained in the largely insoluble mineral apatite (francolite), found in the sedimentary rocks underlying large parts of the Arabian Peninsula (Notholt et al., 2005).

The sum of chemical species, including elements expressed as oxides, and ion concentrations, vary from 35–78 %, with an average of 56 % of the measured chemical mass.  The shortfall from 100
15    % is attributed in part to components not analyzed for, including $H_2O$, OH, carbon ($CO_3^{2-}$, organic carbon, elemental carbon) and artifacts of debris deposited onto the samplers.

The chemical abundances were recalculated as normative minerals (Fig. 12a, b), comparable in composition to those identified by XRD (Fig. 8) and optical microscopy. The relative normative mineral abundances (Fig. 12b) show variable amounts of quartz (avg. 52.4 %) feldspar (avg. 3.9
20    %), kaolinite (2.6 %), calcite (8.8 %) dolomite (0.2 %), hematite (8.0 %), as well as the evaporate minerals gypsum (12.1 %), halite (12.1 %), sylvite (0.2 %), and bischofite (0.2 %). There is also, as shown by XRD, an increase in halite content from about 7.8 % in January to about 25.9 % in July, followed by a sharp drop to about 4.6 % in August, with greater abundances in September (51.0 %) and October (31.6 %) at the CMOR quayside site, ascribed to sea spray from stormy
25    conditions during those two months (Fig. 12b).

Elemental mass ratios of the Frisbee deposition samples are compared to the <38 μm sieved soil samples from the Arabian Red Sea coastal plain (Prakash et al., 2016), and total suspended particulate (TSP) samples collected at other sites in the Middle East (Engelbrecht et al., 2009b) are compared in Table 4. The average Si/Al ratio of 6.86 of the Frisbee deposition samplers is



intermediate to the 13.60 of the Arabian Red Sea coastal soils and the approximately unity of the Middle East samples. The Fe/Al ratios of the sample sets show similar relationships as the Si/Al ratios, being intermediate to the Red Sea coastal soils and four of the five other Middle East countries, excluding UAE, to which it is similar. The difference is ascribed to the greater

abundance of the minerals such as quartz in the coarser sieved soil samples, and less thereof in the finer TSP fractions. The Ca/Al ratio of 2.17 is similar to those of TSP samples from samples of Qatar (2.07) and UAE (2.16), ascribed to the regional carbonate-bearing soils in all three countries. The average Ti/Al, Mg/Al and K/Al ratios of the Frisbee deposition samples are substantially lower than those of the Red Sea coastal soils, which may be related to mineralogical differences in the

dust source regions. Differences can also be ascribed to larger percentages of Al bearing minerals such as clays in the deposition samples from this study.

## 5.    Summary and conclusions

This paper has as its goal the provision of mineralogical, physical and chemical information on deposition samples collected at the KAUST campus during 2015. The study provides an

assessment of the seasonal variability of the regional dust deposition rates onto Saudi Arabian coastal plain, and is meant to be used for validating dust mass balance in the meteorological models with the dust component.

Inverted Frisbee samplers with foam inserts are found to be robust, easy to use, and provided comparable results, for the collection of wet and dry deposits. Once a month the samples are

retrieved by flushing the deposits into plastic flasks followed by freeze-drying of the slurry and recovery of all suspended particles and dissolved salts. The average deposition rate at KAUST in 2015 was 14 g m$^{-2}$, fluctuating from 4 g m$^{-2}$ in December, to 20 g m$^{-2}$ in March, 5 g m$^{-2}$ in July, 28 g m$^{-2}$ in September and October, and down to 11 g m$^{-2}$ the following December. The fluctuations are ascribed to variable meteorological conditions, including high humidity prevailing along the

Arabian Red Sea coastal plain during the autumn months. The particle size distribution patterns point to a marginal bias towards the coarser size fractions in comparison with the <38 μm sieved soil samples.

Chemical analysis, confirmed by XRD, point to a consistent silicate mineral fractions for the deposition samples, at all sampling sites for the entire sampling period.  The Si/Al, Fe/Al, and



Ca/Al ratios of the deposition samples fall within the range of the soil samples previously collected along the Arabian Red Sea coastal plain as well as the TSP size fractions collected at several sites in the Middle East. It is therefore not feasible to explicitly relate the deposition samples to the coastal soils from chemical and mineralogical results on their own. It is proposed that the dust

deposits along the Red Sea coast are a mixture of dust emissions from local soils, and those soils imported from distal dust sources.

For 2015, there are marked similarities between monthly distribution patterns of the deposition samples and AOD measured at KAUST, as well as visibility measurements from Jeddah airport, 70 km to the south. This shows that both the AOD and visibility measurements mirror fluctuations

in dust deposition, although it may not be justified to calculate quantitative interrelationships without further research.

Except for the variable halite fractions, there are small variations in the mineralogical content of the dust samples collected on the KAUST campus. To better represent the dust being deposited in Red Sea, and coastal plain, the sampling campaign should be extended to sites beyond the KAUST

campus. Such a sampling site was recently set up on an island off the coast from KAUST.

## 6. Data availability

The gravimetric, mineralogical and chemical data from this study are available upon request from Georgiy Stenchikov (Georgiy.Stenchikov@kaust.edu.sa).

*Author Contributions*. Johann Engelbrecht was responsible for the sample analysis and data

compilation; Georgiy Stenchikov formulated the problem, designed the research project, and supported experimental activities; Jish Prakash collected, and conducted the freeze-drying of the samples, and performed part of the XRD analysis. Anatolii Anisimov assembled the meteorological and visibility data; Illia Shevchenko assembled the AERONET optical data; Engelbrecht, Stenchikov, Jish Prakash, Anisimov, and Shevchenko compiled different parts of

the manuscript.

*Acknowledgements.* This research, including the chemical and mineralogical analysis is supported by internal funding from the King Abdullah University of Science and Technology (KAUST).  We acknowledge the contributions from the collaborating Core Labs at KAUST and





the Desert Research Institute. This research is supported by the Supercomputing Laboratory at KAUST.

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



**FIGURES**

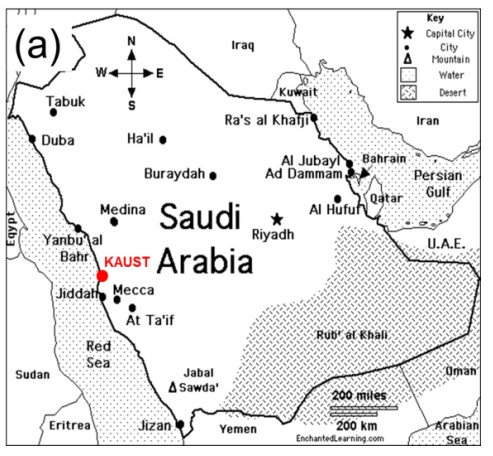
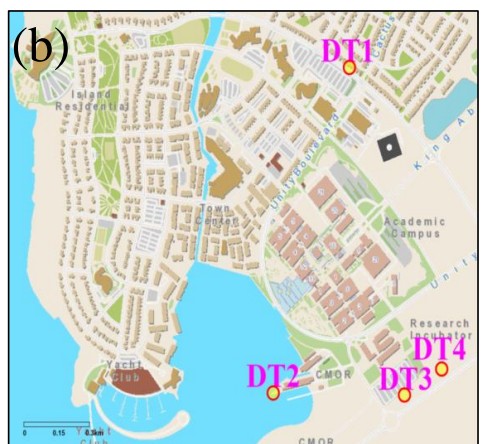

Figure 1. Locality of (a) the King Abdullah University of Science and Technology (KAUST) campus, north of the coastal city of Jeddah, along the Saudi Arabian Red Sea coast and (b) the Frisbee deposition sites (DT1-DT4) on the KAUST campus.





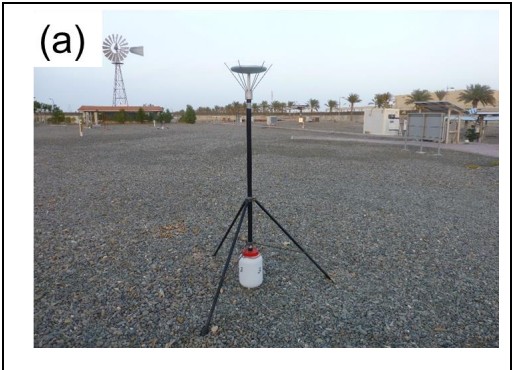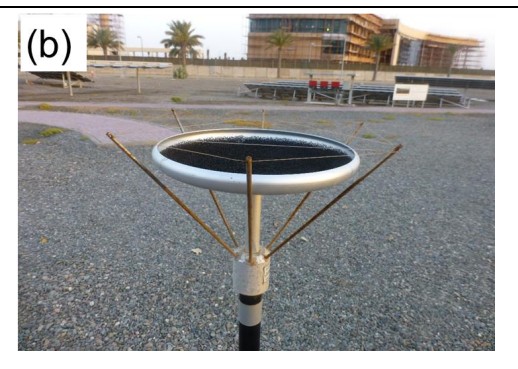

Figure 2. Inverted Frisbee type deposition sampler (a) on tripod and white plastic drainage bottle,

(b) showing the foam insert in the collection dish to help retain the deposited dust particles, as well

5    as the spikes with nylon thread to prevent birds from readily perching on the dish.



Figure 3. Wind roses show prevailing north-westerly *Shamal* winds at KAUST for all months of the 2015 year with north-easterly winds (windspeed in m s[-1]) in the winter months of October through February.





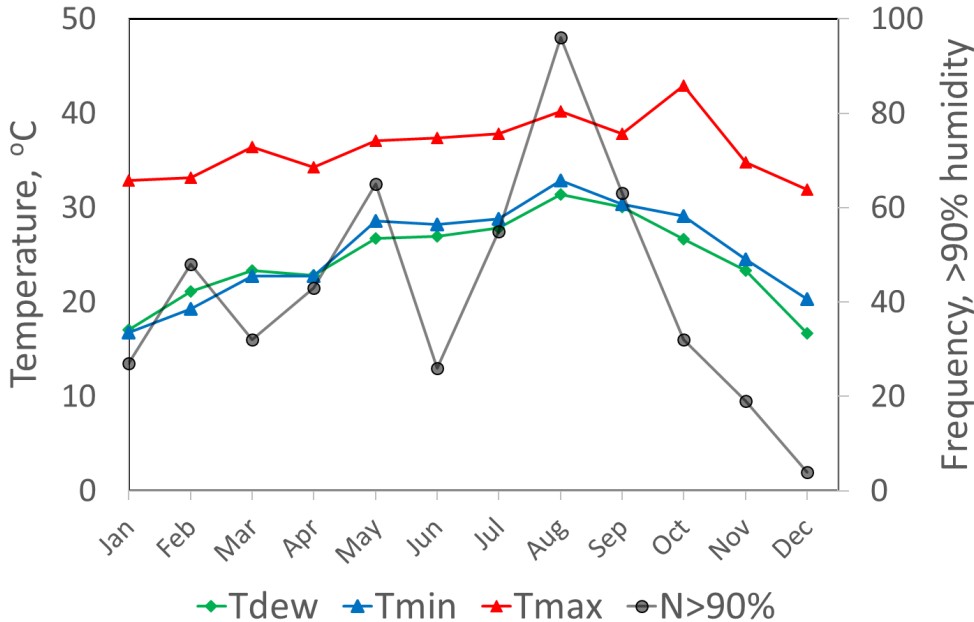

Figure 4. Monthly averaged minimum and maximum ambient temperatures as well as dewpoint variations for KAUST during 2015. Also shown for each month is the frequency of hourly humidity measurements exceeding 90%.



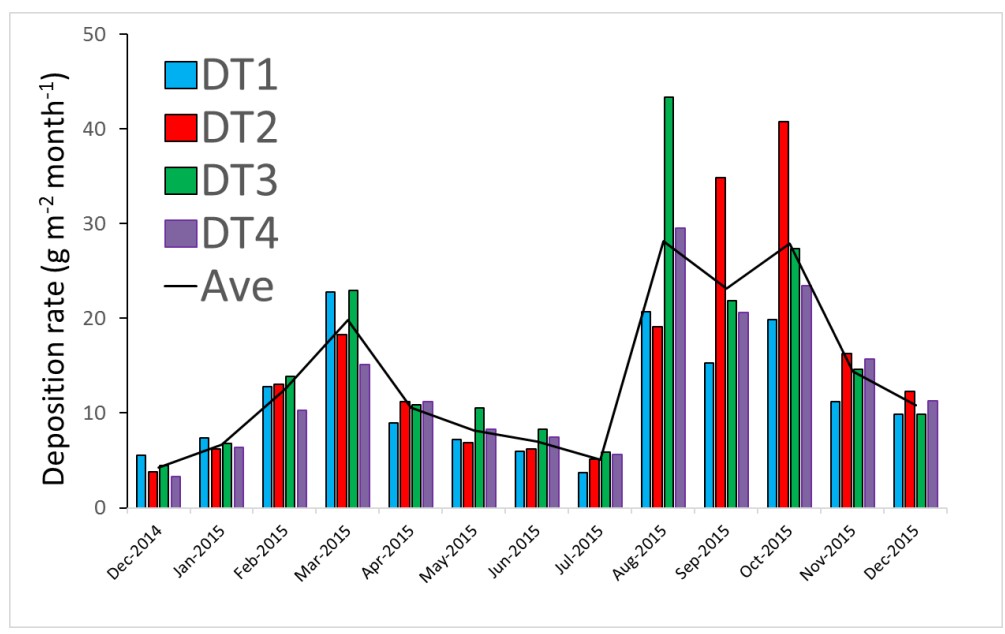

Figure 5. Monthly deposition rates (g m$^{-2}$) from Frisbee samplers (DT1-DT4) at the KAUST
campus. Also shown are the monthly averages for the four samplers.





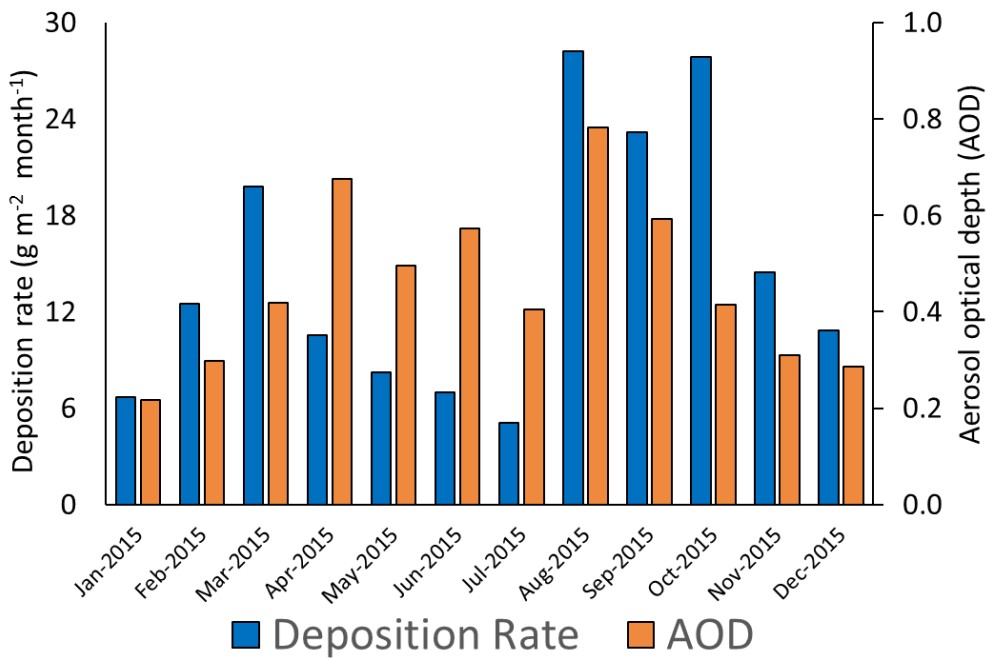

Figure 6. Average monthly deposition rate for all four samplers (DT1- DT4) on the KAUST
campus, and the monthly averaged AOD measurements from the KAUST AERONET site, for
2015.





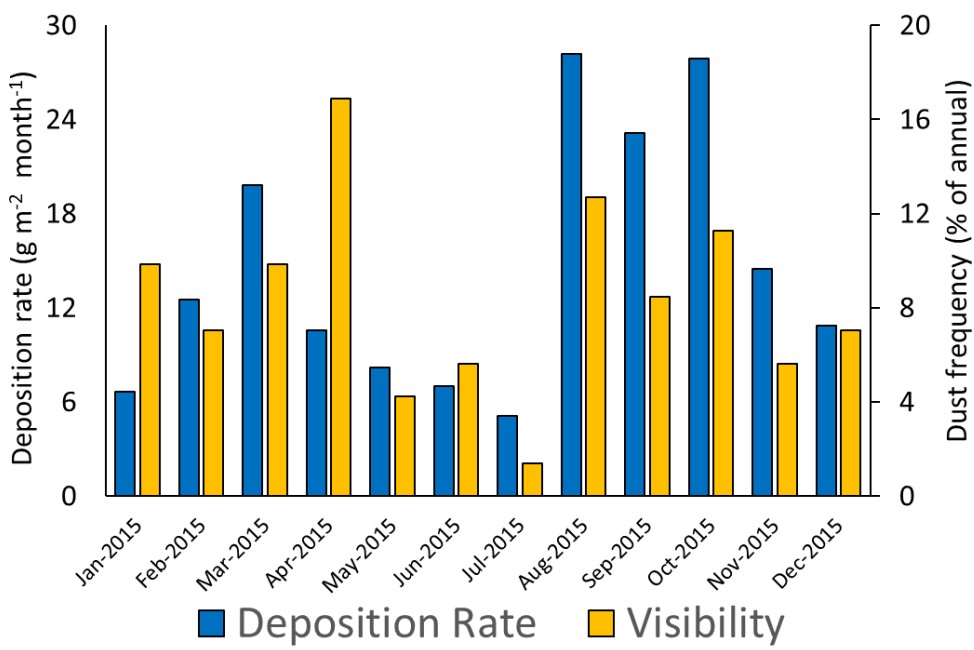

Figure 7. Average monthly deposition rate for all four samplers (DT1-DT4) on the KAUST
campus, and the monthly averaged visibility measurements collected from the Jeddah airport in
2015.





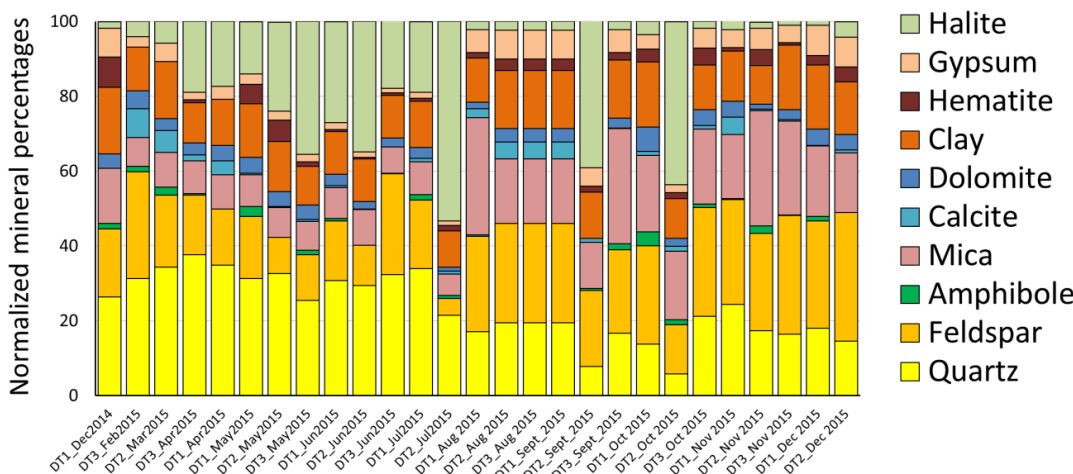

Figure 8. Semi-quantitative XRD mineral analyses of monthly Frisbee samples collected at the

5    three sites DT1–DT3, for the period December, 2014 to December2015.



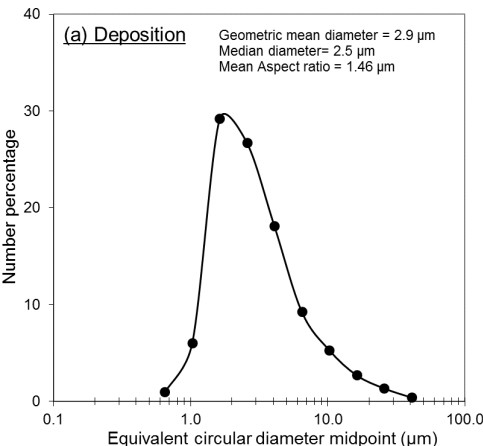
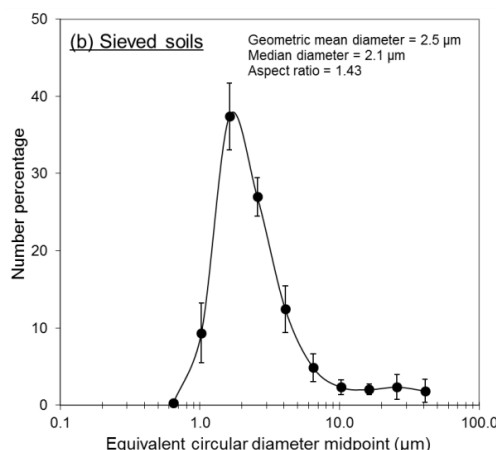

Figure 9. Particle size distributions of (a) deposition sample (DT1) collected by Frisbee sampler on KAUST campus, and (b) average of thirteen <38 μm sieved soil samples (Prakash et al., 2016), both measured by SEM.



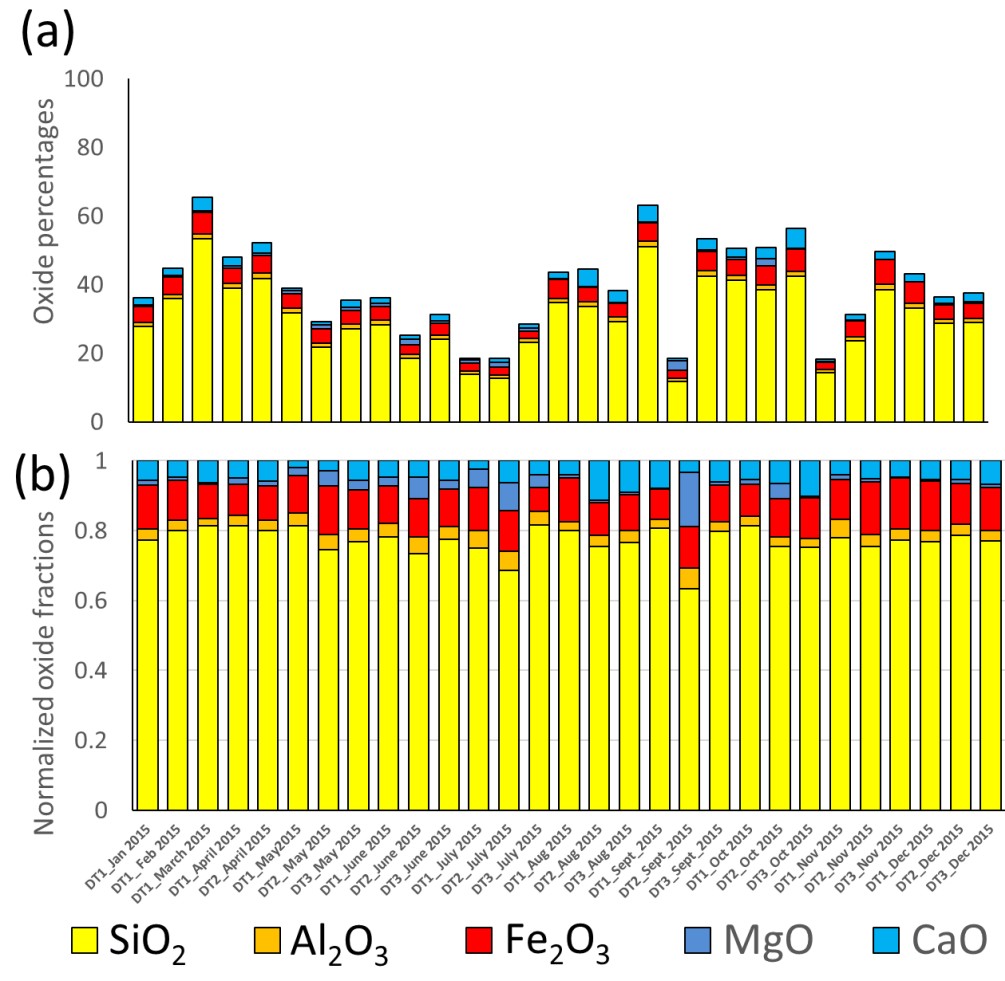

Figure 10(a). Deposition sample elemental compositions, expressed as oxides and (b) fractions
5    normalized to unity.





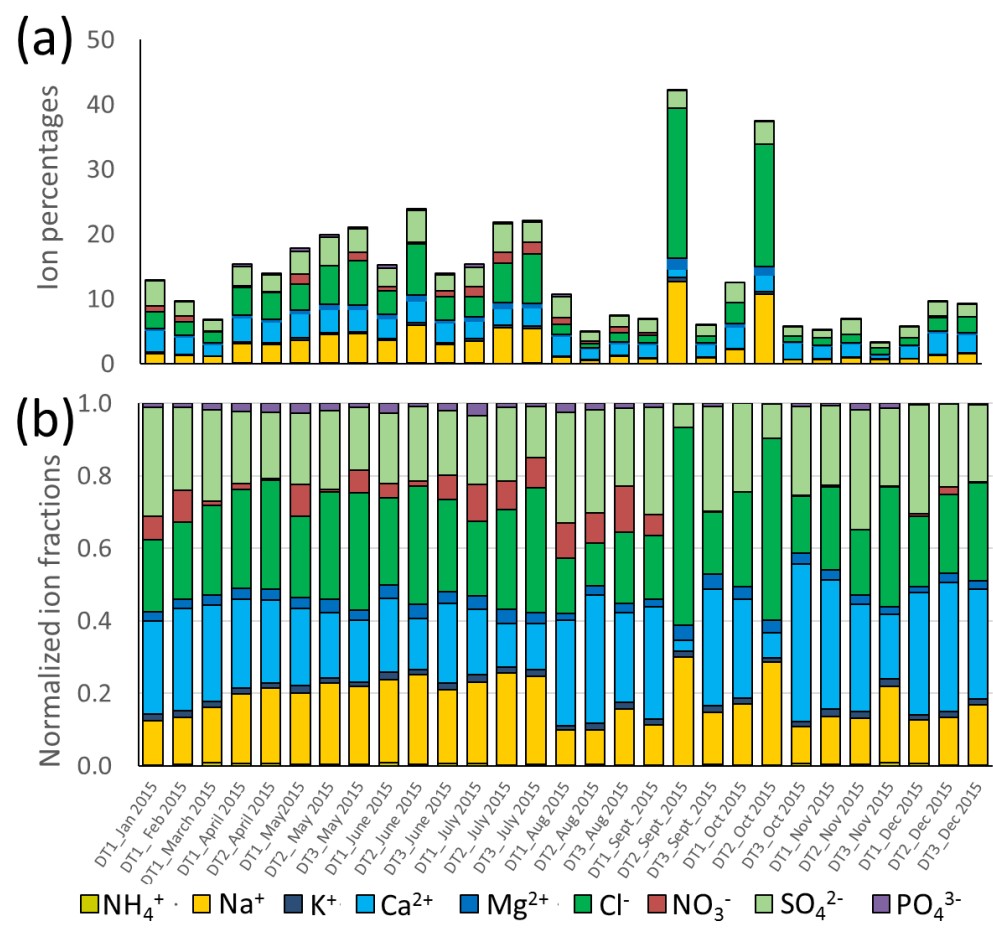

Figure 11 (a). Ion concentrations and (b) fractions totaled to unity.



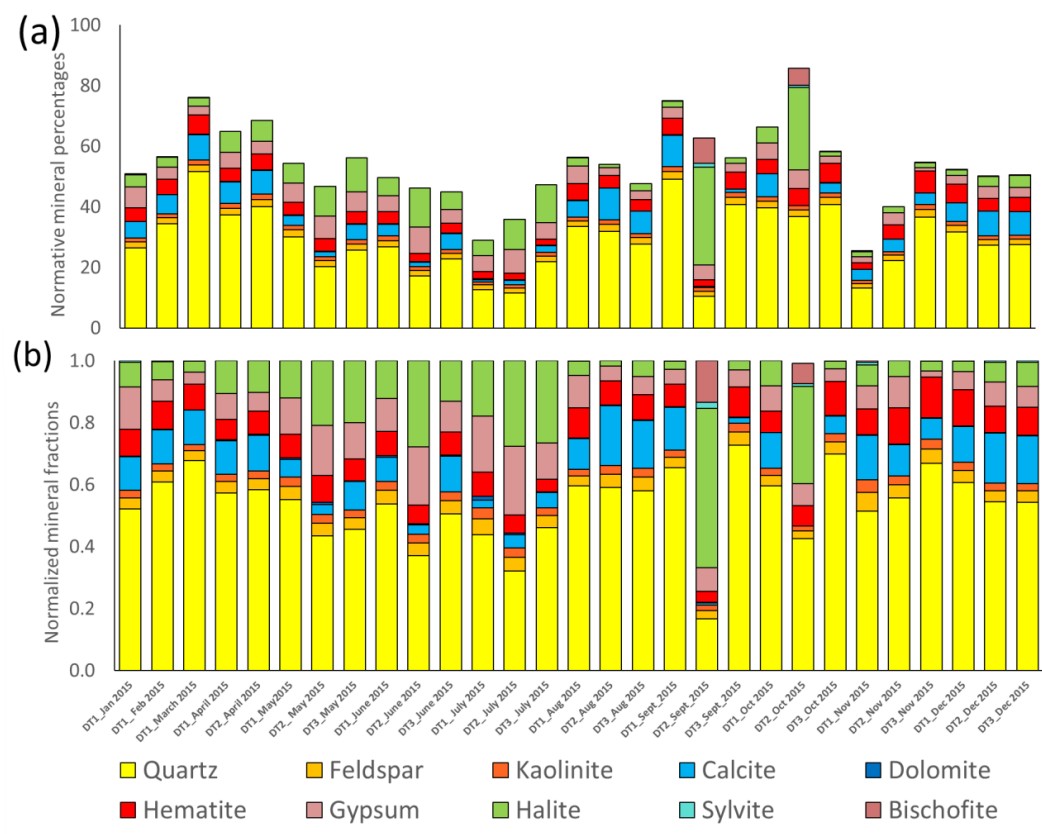

Figure 12 (a). Chemical abundances combined as normative minerals, (b) normalized to 100%.



## TABLES

Table 1. Locality of deposition samplers at six sites on the campus of KAUST.

|  | Site | Latitude | Longitude | Elev. m.a.s.l. | Start | End |
|---|---|---|---|---|---|---|
| DT1 | NEO 1 | 22°18'16.12"N | 39°06'28.46"E | 1 | Dec14 | Mar15 |
|  | Res G3705 | 22°18'59.06"N | 39°06'21.32"E | 12 | Apr15 | Dec15 |
| DT2 | NEO 2 | 22°18'16.84"N | 39° 6'29.33"E | 1 | Dec14 | Mar15 |
|  | CMOR | 22°18'16.60"N | 39° 6'7.91"E | 1 | Apr15 | Dec15 |
| DT3 | NEO 3 | 22°18'17.31"N | 39° 6'30.51"E | 1 | Dec14 | Dec15 |
| DT4 | NEO 4 | 22°18'18.10"N | 39° 6'31.52"E | 1 | Dec14 | Dec15 |





Table 2. Monthly averaged temperatures, humidity measurements, and calculated dewpoints at KAUST during 2015.

| Month | Temperature | | | | Humidity | | | | Dewpoint |
|---|---|---|---|---|---|---|---|---|---|
| | Avg. °C | Min. °C | Max. °C | Range °C | Avg. % | Min. % | Max. % | N>90% Count | Avg. °C |
| Jan | 25 | 17 | 33 | 16 | 61 | 10 | 99 | 27 | 17 |
| Feb | 27 | 19 | 33 | 14 | 74 | 15 | 99 | 48 | 21 |
| Mar | 28 | 23 | 36 | 14 | 76 | 29 | 99 | 32 | 23 |
| Apr | 28 | 23 | 34 | 12 | 74 | 23 | 99 | 43 | 23 |
| May | 32 | 29 | 37 | 9 | 77 | 21 | 99 | 65 | 27 |
| Jun | 32 | 28 | 37 | 9 | 76 | 22 | 99 | 26 | 27 |
| Jul | 33 | 29 | 38 | 9 | 75 | 26 | 99 | 55 | 28 |
| Aug | 35 | 33 | 40 | 7 | 82 | 36 | 99 | 96 | 31 |
| Sep | 34 | 30 | 38 | 8 | 80 | 26 | 99 | 63 | 30 |
| Oct | 33 | 29 | 43 | 14 | 72 | 9 | 96 | 32 | 27 |
| Nov | 30 | 25 | 35 | 10 | 69 | 25 | 99 | 19 | 23 |
| Dec | 27 | 20 | 32 | 12 | 57 | 15 | 94 | 4 | 17 |



Table 3. Dust deposition measurements from the Middle East and other Global dust regions.

| | Study | Locality | Sampler type | Sampling period | Average deposition rate (g m$^{-2}$ month$^{-1}$) | Range deposition rate (g m$^{-2}$ month$^{-1}$) |
|---|---|---|---|---|---|---|
| (a) | This study (2017) | Saudi Arabia, KAUST | Frisbee with foam insert | Dec 2014 - Dec 2015 | 14 | 4 - 28 |
| (b) | Modaihsh and Mahjoub (2013) | Saudi Arabia, Riyadh | Dish with marbles | Jan - Mar ? | 42 | 20 - 140 |
| (c) | Khalaf and Al-Hashash (1983) | Kuwait, N-W Gulf | Polyethelene cylinders with water | Apr 1979 - Mar 1980 | 191 | 10 - 1003 |
| (d) | Al-Awadhi (2005) | Kuwait, N-E Bay | PVC bucket with marbles | May 2002 - Apr 2003 | 28 | 3 - 58 |
| (e) | Al-Awadhi and AlShuaibi (2013) | Kuwait, City | PVC bucket with marbles | Mar 2011 - Feb 2012 | 53 | 2 - 320 |
| (f) | Offer and Goossens (2001) | Israel, Negev | Marble collectors | 1988 - 1997 | 17 | 10 - 25 |
| (g) | Goossens and Rajot (2008) | Niger, Banizoumbou | Frisbee with marbles, original data | 8 periods in 2005 | 13 | 6 - 21 |
| (h) | Smith and Twiss (1965) | USA, Kansas | Cylindrical rain gauge with screens | June 1963 - June 1964 | 6 | 3 - 14 |





Table 4. Elemental mass ratios for the deposition samples from this study, compared to those of soils from the Red Sea coastal plain (Prakash et al., 2016) and TSP samples from other countries
5   of the Middle East (Engelbrecht et al., 2009b). The TSP filter samples were collected by low-volume aerosol samplers without size selective inlets, for 24-hr sampling periods.

|  |  | Si/Al | Ti/Al | Fe/Al | Mg/Al | Ca/Al | K/Al |
|---|---|---|---|---|---|---|---|
| Frisbee | Deposition | 6.86 | 0.14 | 1.47 | 0.11 | 2.17 | 0.34 |
| Saudi Soils | Sieved <38µm | 13.60 | 0.44 | 2.52 | 0.65 | 0.36 | 0.43 |
| Djibouti | TSP | 0.92 | 2.19 | 1.12 | 0.88 | 0.74 | 1.14 |
| Afghanistan | TSP | 1.05 | 1.25 | 1.00 | 0.94 | 0.69 | 1.96 |
| Qatar | TSP | 1.02 | 0.24 | 0.98 | 1.40 | 2.07 | 0.93 |
| UAE | TSP | 1.29 | 0.28 | 1.52 | 2.85 | 2.16 | 1.02 |
| Iraq | TSP | 1.03 | 0.72 | 0.99 | 1.11 | 1.31 | 1.04 |
| Kuwait | TSP | 1.07 | 0.65 | 0.99 | 1.25 | 1.23 | 0.94 |




Appendix A

| Sample | DT1_Jan 2015 | | DT1_ Feb 2015 | | DT1_March 2015 | | DT1_April 2015 | | DT2_April 2015 | | DT1_May2015 | | DT2_ May 2015 | |
|---|---|---|---|---|---|---|---|---|---|---|---|---|---|---|
| **Major and minor elements as oxides (%)** | | | | | | | | | | | | | | |
| $SiO_2$ | 27.890 | ± 0.050 | 35.886 | ± 0.065 | 53.301 | ± 0.089 | 38.965 | ± 0.063 | 41.802 | ± 0.073 | 31.729 | ± 0.055 | 21.772 | ± 0.039 |
| $TiO_2$ | 0.466 | ± 0.001 | 0.530 | ± 0.001 | 0.679 | ± 0.002 | 0.464 | ± 0.002 | 0.599 | ± 0.002 | 0.486 | ± 0.002 | 0.451 | ± 0.001 |
| $Al_2O_3$ | 3.505 | ± 0.035 | 4.464 | ± 0.050 | 5.421 | ± 0.097 | 4.257 | ± 0.115 | 4.824 | ± 0.081 | 3.817 | ± 0.079 | 3.033 | ± 0.026 |
| $Fe_2O_3$ | 4.480 | ± 0.007 | 5.119 | ± 0.008 | 6.312 | ± 0.013 | 4.309 | ± 0.013 | 5.091 | ± 0.011 | 4.150 | ± 0.010 | 4.104 | ± 0.006 |
| MnO | 0.080 | ± 0.001 | 0.082 | ± 0.002 | 0.109 | ± 0.004 | 0.075 | ± 0.005 | 0.091 | ± 0.003 | 0.077 | ± 0.003 | 0.066 | ± 0.001 |
| CaO* | 1.989 | ± 0.015 | 2.097 | ± 0.013 | 4.157 | ± 0.016 | 2.369 | ± 0.018 | 3.066 | ± 0.017 | 0.753 | ± 0.014 | 0.847 | ± 0.013 |
| $K_2O$* | 0.484 | ± 0.005 | 0.604 | ± 0.004 | 0.864 | ± 0.003 | 0.705 | ± 0.006 | 0.798 | ± 0.005 | 0.542 | ± 0.008 | 0.432 | ± 0.007 |
| $P_2O_5$ | 0.417 | ± 0.001 | 0.150 | ± 0.001 | 0.010 | ± 0.003 | 0.332 | ± 0.004 | 0.306 | ± 0.002 | 0.474 | ± 0.003 | 0.429 | ± 0.001 |
| **Total (oxides)** | 39.311 | | 48.932 | | 70.852 | | 51.475 | | 56.579 | | 42.028 | | 31.134 | |
| | | | | | | | | | | | | | | |
| **Trace elements (ppm)** | | | | | | | | | | | | | | |
| V | 107 | ± 1 | 76 | ± 1 | 85 | ± 3 | 106 | ± 4 | 93 | ± 3 | 116 | ± 3 | 99 | ± 1 |
| Cr | 92 | ± 3 | 115 | ± 5 | 113 | ± 12 | 101 | ± 15 | 108 | ± 10 | 113 | ± 10 | 120 | ± 2 |
| Ni | 78 | ± 2 | 71 | ± 3 | 84 | ± 6 | 68 | ± 8 | 71 | ± 5 | 77 | ± 5 | 57 | ± 1 |
| Cu | 81 | ± 3 | 57 | ± 4 | 134 | ± 10 | 125 | ± 13 | 99 | ± 9 | 206 | ± 9 | 64 | ± 2 |
| Zn | 223 | ± 5 | 247 | ± 8 | 293 | ± 18 | 287 | ± 23 | 258 | ± 15 | 467 | ± 16 | 219 | ± 4 |
| As | 0 | ± 3 | 0 | ± 5 | 6 | ± 12 | 0 | ± 15 | 0 | ± 10 | 0 | ± 10 | 0 | ± 2 |
| Br | 29 | ± 3 | 26 | ± 4 | 42 | ± 10 | 59 | ± 13 | 45 | ± 9 | 34 | ± 9 | 76 | ± 2 |
| Rb | 28 | ± 1 | 24 | ± 1 | 16 | ± 3 | 28 | ± 4 | 34 | ± 3 | 30 | ± 3 | 27 | ± 1 |
| Sr | 333 | ± 3 | 392 | ± 5 | 514 | ± 11 | 386 | ± 13 | 422 | ± 9 | 303 | ± 9 | 341 | ± 2 |
| Y | 1071 | ± 4 | 36 | ± 4 | 26 | ± 10 | 125 | ± 13 | 46 | ± 9 | 36 | ± 9 | 24 | ± 2 |
| Zr | 103 | ± 4 | 98 | ± 6 | 84 | ± 14 | 57 | ± 17 | 139 | ± 12 | 133 | ± 12 | 134 | ± 3 |
| Mo | 0 | ± 5 | 0 | ± 7 | 12 | ± 17 | 4 | ± 21 | 8 | ± 14 | 0 | ± 14 | 0 | ± 3 |
| Pb | 15 | ± 5 | 28 | ± 8 | 32 | ± 18 | 66 | ± 23 | 28 | ± 15 | 19 | ± 15 | 28 | ± 3 |
| | | | | | | | | | | | | | | |
| **Water soluble ions (%)** | | | | | | | | | | | | | | |
| $NH_4^+$ | 0.027 | ± 0.003 | 0.033 | ± 0.004 | 0.055 | ± 0.006 | 0.105 | ± 0.012 | 0.095 | ± 0.011 | 0.076 | ± 0.009 | 0.079 | ± 0.009 |
| $Na^+$ | 1.585 | ± 0.012 | 1.267 | ± 0.010 | 1.053 | ± 0.009 | 2.952 | ± 0.021 | 2.897 | ± 0.021 | 3.519 | ± 0.025 | 4.473 | ± 0.031 |
| $K^+$ | 0.230 | ± 0.009 | 0.170 | ± 0.006 | 0.114 | ± 0.004 | 0.262 | ± 0.010 | 0.207 | ± 0.008 | 0.371 | ± 0.014 | 0.296 | ± 0.011 |
| $Mg^{2+}$ | 0.338 | ± 0.005 | 0.265 | ± 0.004 | 0.188 | ± 0.003 | 0.488 | ± 0.007 | 0.400 | ± 0.005 | 0.558 | ± 0.008 | 0.744 | ± 0.010 |
| $Ca^{2+}$ | 3.311 | ± 0.028 | 2.718 | ± 0.023 | 1.813 | ± 0.015 | 3.745 | ± 0.032 | 3.220 | ± 0.027 | 3.780 | ± 0.032 | 3.560 | ± 0.030 |
| $Cl^-$ | 2.563 | ± 0.014 | 2.054 | ± 0.011 | 1.695 | ± 0.009 | 4.177 | ± 0.022 | 4.227 | ± 0.023 | 3.963 | ± 0.021 | 5.927 | ± 0.032 |
| $NO_3^-$ | 0.830 | ± 0.030 | 0.843 | ± 0.031 | 0.082 | ± 0.004 | 0.269 | ± 0.010 | 0.051 | ± 0.004 | 1.578 | ± 0.057 | 0.107 | ± 0.005 |
| $SO_4^{2-}$ | 3.877 | ± 0.034 | 2.199 | ± 0.019 | 1.718 | ± 0.015 | 3.055 | ± 0.027 | 2.549 | ± 0.022 | 3.527 | ± 0.031 | 4.327 | ± 0.038 |
| **Total (ions)** | 12.761 | | 9.548 | | 6.719 | | 15.052 | | 13.647 | | 17.372 | | 19.513 | |
| **Total (oxides + ions)** | 52.072 | | 58.479 | | 77.571 | | 66.527 | | 70.226 | | 59.400 | | 50.647 | |
| **Note: CaO* and K2O* are water insoluble, $P_2O_5^-$ calculated from total P** | | | | | | | | | | | | | | |



# Appendix A

| Sample | DT3_May 2015 | | | DT1_June 2015 | | | DT2_June 2015 | | | DT3_June 2015 | | | DT1_July 2015 | | | DT2_July 2015 | | | DT3_July 2015 | | |
|---|---|---|---|---|---|---|---|---|---|---|---|---|---|---|---|---|---|---|---|---|---|
| **Major and minor elements as oxides (%)** | | | | | | | | | | | | | | | | | | | | | |
| $SiO_2$ | 27.167 | ± | 0.051 | 28.318 | ± | 0.053 | 18.530 | ± | 0.035 | 24.158 | ± | 0.045 | 13.820 | ± | 0.027 | 12.693 | ± | 0.025 | 23.221 | ± | 0.037 |
| $TiO_2$ | 0.448 | ± | 0.001 | 0.478 | ± | 0.002 | 0.306 | ± | 0.001 | 0.370 | ± | 0.001 | 0.261 | ± | 0.001 | 0.241 | ± | 0.001 | 0.200 | ± | 0.002 |
| $Al_2O_3$ | 3.777 | ± | 0.059 | 3.692 | ± | 0.062 | 2.919 | ± | 0.050 | 3.067 | ± | 0.045 | 1.886 | ± | 0.041 | 2.030 | ± | 0.037 | 2.694 | ± | 0.148 |
| $Fe_2O_3$ | 3.972 | ± | 0.008 | 3.906 | ± | 0.008 | 2.782 | ± | 0.006 | 3.346 | ± | 0.006 | 2.287 | ± | 0.005 | 2.154 | ± | 0.005 | 1.935 | ± | 0.014 |
| MnO | 0.083 | ± | 0.002 | 0.077 | ± | 0.002 | 0.050 | ± | 0.002 | 0.068 | ± | 0.002 | 0.050 | ± | 0.002 | 0.044 | ± | 0.002 | 0.050 | ± | 0.007 |
| CaO* | 1.983 | ± | 0.016 | 1.681 | ± | 0.014 | 1.179 | ± | 0.013 | 1.745 | ± | 0.014 | 0.427 | ± | 0.010 | 1.179 | ± | 0.011 | 1.148 | ± | 0.013 |
| $K_2O$* | 0.508 | ± | 0.006 | 0.513 | ± | 0.007 | 0.366 | ± | 0.008 | 0.467 | ± | 0.006 | 0.208 | ± | 0.008 | 0.226 | ± | 0.008 | 0.359 | ± | 0.009 |
| $P_2O_5$ | 0.170 | ± | 0.002 | 0.590 | ± | 0.002 | 0.383 | ± | 0.002 | 0.389 | ± | 0.001 | 0.753 | ± | 0.002 | 0.457 | ± | 0.001 | 0.288 | ± | 0.005 |
| **Total (oxides)** | 38.108 | | | 39.254 | | | 26.514 | | | 33.609 | | | 19.692 | | | 19.024 | | | 29.895 | | |
| | | | | | | | | | | | | | | | | | | | | | |
| **Trace elements (ppm)** | | | | | | | | | | | | | | | | | | | | | |
| V | 88 | ± | 2 | 111 | ± | 2 | 86 | ± | 2 | 91 | ± | 1 | 115 | ± | 1 | 111 | ± | 1 | 77 | ± | 5 |
| Cr | 71 | ± | 7 | 96 | ± | 7 | 65 | ± | 6 | 82 | ± | 5 | 160 | ± | 5 | 60 | ± | 5 | 67 | ± | 21 |
| Ni | 62 | ± | 3 | 84 | ± | 4 | 49 | ± | 3 | 72 | ± | 3 | 133 | ± | 3 | 71 | ± | 2 | 91 | ± | 11 |
| Cu | 56 | ± | 6 | 134 | ± | 7 | 59 | ± | 5 | 49 | ± | 5 | 209 | ± | 5 | 62 | ± | 4 | 25 | ± | 19 |
| Zn | 251 | ± | 11 | 430 | ± | 12 | 297 | ± | 9 | 255 | ± | 8 | 515 | ± | 9 | 244 | ± | 7 | 180 | ± | 32 |
| As | 0 | ± | 7 | 0 | ± | 7 | 0 | ± | 6 | 0 | ± | 5 | 0 | ± | 5 | 0 | ± | 5 | 0 | ± | 21 |
| Br | 58 | ± | 6 | 62 | ± | 7 | 100 | ± | 5 | 64 | ± | 5 | 62 | ± | 5 | 88 | ± | 4 | 37 | ± | 19 |
| Rb | 24 | ± | 2 | 35 | ± | 2 | 16 | ± | 2 | 23 | ± | 1 | 17 | ± | 1 | 12 | ± | 1 | 8 | ± | 5 |
| Sr | 392 | ± | 7 | 322 | ± | 7 | 276 | ± | 6 | 291 | ± | 5 | 233 | ± | 5 | 268 | ± | 4 | 284 | ± | 19 |
| Y | 14 | ± | 6 | 23 | ± | 7 | 14 | ± | 5 | 19 | ± | 5 | 7 | ± | 5 | 12 | ± | 4 | 23 | ± | 19 |
| Zr | 123 | ± | 8 | 149 | ± | 9 | 112 | ± | 7 | 82 | ± | 6 | 76 | ± | 6 | 83 | ± | 6 | 88 | ± | 24 |
| Mo | 0 | ± | 10 | 0 | ± | 10 | 0 | ± | 8 | 0 | ± | 7 | 0 | ± | 7 | 0 | ± | 6 | 0 | ± | 29 |
| Pb | 18 | ± | 10 | 12 | ± | 11 | 15 | ± | 9 | 17 | ± | 8 | 22 | ± | 8 | 30 | ± | 7 | 45 | ± | 32 |
| | | | | | | | | | | | | | | | | | | | | | |
| **Water soluble ions (%)** | | | | | | | | | | | | | | | | | | | | | |
| $NH_4^+$ | 0.091 | ± | 0.010 | 0.125 | ± | 0.014 | 0.120 | ± | 0.013 | 0.100 | ± | 0.011 | 0.088 | ± | 0.010 | 0.032 | ± | 0.004 | 0.109 | ± | 0.012 |
| $Na^+$ | 4.509 | ± | 0.031 | 3.505 | ± | 0.025 | 5.875 | ± | 0.041 | 2.834 | ± | 0.020 | 3.466 | ± | 0.024 | 5.538 | ± | 0.038 | 5.360 | ± | 0.037 |
| $K^+$ | 0.276 | ± | 0.010 | 0.312 | ± | 0.012 | 0.352 | ± | 0.013 | 0.256 | ± | 0.010 | 0.339 | ± | 0.013 | 0.360 | ± | 0.013 | 0.390 | ± | 0.015 |
| $Mg^{2+}$ | 0.572 | ± | 0.008 | 0.559 | ± | 0.008 | 0.947 | ± | 0.013 | 0.453 | ± | 0.006 | 0.582 | ± | 0.008 | 0.878 | ± | 0.012 | 0.628 | ± | 0.009 |
| $Ca^{2+}$ | 3.582 | ± | 0.030 | 3.094 | ± | 0.026 | 3.341 | ± | 0.028 | 3.092 | ± | 0.026 | 2.767 | ± | 0.024 | 2.642 | ± | 0.022 | 2.833 | ± | 0.024 |
| $Cl^-$ | 6.828 | ± | 0.037 | 3.648 | ± | 0.020 | 7.799 | ± | 0.042 | 3.562 | ± | 0.019 | 3.141 | ± | 0.017 | 6.005 | ± | 0.032 | 7.612 | ± | 0.041 |
| $NO_3^-$ | 1.328 | ± | 0.048 | 0.595 | ± | 0.022 | 0.349 | ± | 0.013 | 0.939 | ± | 0.034 | 1.573 | ± | 0.057 | 1.702 | ± | 0.061 | 1.864 | ± | 0.067 |
| $SO_4^{2-}$ | 3.649 | ± | 0.032 | 2.965 | ± | 0.026 | 4.878 | ± | 0.043 | 2.483 | ± | 0.022 | 2.923 | ± | 0.026 | 4.424 | ± | 0.039 | 3.087 | ± | 0.027 |
| **Total (ions)** | 20.836 | | | 14.804 | | | 23.663 | | | 13.719 | | | 14.879 | | | 21.580 | | | 21.882 | | |
| **Total (oxides + ions)** | 58.944 | | | 54.058 | | | 50.177 | | | 47.328 | | | 34.571 | | | 40.605 | | | 51.777 | | |

Note: CaO* and K2O* are water insoluble, $P_2O_5^-$ calculated from total P





## Appendix A

| Sample | DT1_Aug 2015 | | | DT2_Aug 2015 | | | DT3_Aug 2015 | | | DT1_Sept_2015 | | | DT2_Sept_2015 | | | DT3_Sept_2015 | | | DT1_Oct 2015 | | |
|---|---|---|---|---|---|---|---|---|---|---|---|---|---|---|---|---|---|---|---|---|---|
| **Major and minor elements as oxides (%)** | | | | | | | | | | | | | | | | | | | | | |
| $SiO_2$ | 34.862 | ± | 0.061 | 33.619 | ± | 0.058 | 29.244 | ± | 0.054 | 50.971 | ± | 0.083 | 11.690 | ± | 0.024 | 42.544 | ± | 0.074 | 41.270 | ± | 0.067 |
| $TiO_2$ | 0.591 | ± | 0.001 | 0.469 | ± | 0.002 | 0.422 | ± | 0.002 | 0.605 | ± | 0.003 | 0.206 | ± | 0.001 | 0.609 | ± | 0.002 | 0.496 | ± | 0.002 |
| $Al_2O_3$ | 4.021 | ± | 0.033 | 3.773 | ± | 0.078 | 3.514 | ± | 0.068 | 5.057 | ± | 0.124 | 3.628 | ± | 0.053 | 5.114 | ± | 0.081 | 4.374 | ± | 0.115 |
| $Fe_2O_3$ | 5.488 | ± | 0.008 | 4.186 | ± | 0.010 | 3.878 | ± | 0.009 | 5.479 | ± | 0.015 | 2.206 | ± | 0.006 | 5.540 | ± | 0.011 | 4.592 | ± | 0.013 |
| $MnO$ | 0.095 | ± | 0.001 | 0.082 | ± | 0.003 | 0.068 | ± | 0.003 | 0.102 | ± | 0.005 | 0.048 | ± | 0.002 | 0.096 | ± | 0.003 | 0.081 | ± | 0.005 |
| $CaO*$ | 1.772 | ± | 0.014 | 5.067 | ± | 0.017 | 3.423 | ± | 0.014 | 4.909 | ± | 0.019 | 3.273 | ± | 0.006 | 0.598 | ± | 0.014 | 2.690 | ± | 0.018 |
| $K_2O*$ | 0.598 | ± | 0.003 | 0.850 | ± | 0.002 | 0.709 | ± | 0.003 | 0.988 | ± | 0.002 | 0.000 | ± | 0.014 | 0.853 | ± | 0.003 | 0.756 | ± | 0.004 |
| $P_2O_5$ | 0.249 | ± | 0.001 | 0.158 | ± | 0.002 | 0.076 | ± | 0.002 | 0.000 | ± | 0.004 | 0.000 | ± | 0.001 | 0.194 | ± | 0.002 | 0.047 | ± | 0.003 |
| **Total (oxides)** | 47.676 | | | 48.204 | | | 41.334 | | | 68.110 | | | 21.051 | | | 55.547 | | | 54.306 | | |
| | | | | | | | | | | | | | | | | | | | | | |
| **Trace elements (ppm)** | | | | | | | | | | | | | | | | | | | | | |
| V | 122 | ± | 1 | 110 | ± | 3 | 42 | ± | 2 | 51 | ± | 4 | 19 | ± | 2 | 93 | ± | 2 | 0 | ± | 4 |
| Cr | 116 | ± | 3 | 59 | ± | 10 | 76 | ± | 9 | 96 | ± | 16 | 46 | ± | 6 | 93 | ± | 10 | 152 | ± | 15 |
| Ni | 91 | ± | 1 | 71 | ± | 5 | 54 | ± | 4 | 61 | ± | 8 | 29 | ± | 3 | 76 | ± | 5 | 73 | ± | 8 |
| Cu | 55 | ± | 2 | 76 | ± | 9 | 41 | ± | 7 | 55 | ± | 14 | 32 | ± | 5 | 86 | ± | 8 | 40 | ± | 13 |
| Zn | 194 | ± | 4 | 226 | ± | 15 | 127 | ± | 13 | 157 | ± | 24 | 107 | ± | 9 | 339 | ± | 15 | 185 | ± | 23 |
| As | 0 | ± | 3 | 0 | ± | 10 | 0 | ± | 9 | 0 | ± | 16 | 0 | ± | 6 | 0 | ± | 10 | 0 | ± | 15 |
| Br | 20 | ± | 2 | 16 | ± | 9 | 34 | ± | 7 | 17 | ± | 14 | 715 | ± | 7 | 58 | ± | 8 | 64 | ± | 13 |
| Rb | 22 | ± | 1 | 43 | ± | 3 | 12 | ± | 2 | 47 | ± | 4 | 18 | ± | 2 | 38 | ± | 2 | 24 | ± | 4 |
| Sr | 441 | ± | 3 | 369 | ± | 9 | 257 | ± | 7 | 394 | ± | 14 | 179 | ± | 5 | 345 | ± | 9 | 331 | ± | 13 |
| Y | 18 | ± | 2 | 9 | ± | 9 | 25 | ± | 7 | 23 | ± | 14 | 8 | ± | 5 | 14 | ± | 8 | 18 | ± | 13 |
| Zr | 141 | ± | 3 | 133 | ± | 12 | 125 | ± | 10 | 123 | ± | 20 | 62 | ± | 7 | 127 | ± | 12 | 99 | ± | 17 |
| Mo | 0 | ± | 4 | 0 | ± | 14 | 0 | ± | 12 | 0 | ± | 23 | 1 | ± | 8 | 3 | ± | 13 | 17 | ± | 21 |
| Pb | 4 | ± | 4 | 58 | ± | 15 | 63 | ± | 13 | 36 | ± | 24 | 12 | ± | 9 | 28 | ± | 14 | 46 | ± | 23 |
| | | | | | | | | | | | | | | | | | | | | | |
| **Water soluble ions (%)** | | | | | | | | | | | | | | | | | | | | | |
| $NH_4^+$ | 0.025 | ± | 0.003 | 0.018 | ± | 0.002 | 0.021 | ± | 0.002 | 0.021 | ± | 0.003 | 0.027 | ± | 0.003 | 0.024 | ± | 0.003 | 0.013 | ± | 0.002 |
| $Na^+$ | 1.044 | ± | 0.009 | 0.481 | ± | 0.006 | 1.144 | ± | 0.009 | 0.756 | ± | 0.007 | 12.685 | ± | 0.087 | 0.862 | ± | 0.008 | 2.134 | ± | 0.016 |
| $K^+$ | 0.116 | ± | 0.004 | 0.094 | ± | 0.004 | 0.133 | ± | 0.005 | 0.106 | ± | 0.004 | 0.625 | ± | 0.023 | 0.123 | ± | 0.005 | 0.192 | ± | 0.007 |
| $Mg^{2+}$ | 0.213 | ± | 0.003 | 0.133 | ± | 0.002 | 0.193 | ± | 0.003 | 0.146 | ± | 0.002 | 1.732 | ± | 0.024 | 0.261 | ± | 0.004 | 0.425 | ± | 0.006 |
| $Ca^{2+}$ | 3.101 | ± | 0.026 | 1.772 | ± | 0.015 | 1.840 | ± | 0.016 | 2.143 | ± | 0.018 | 1.280 | ± | 0.011 | 1.928 | ± | 0.016 | 3.415 | ± | 0.029 |
| $Cl^-$ | 1.624 | ± | 0.009 | 0.588 | ± | 0.004 | 1.460 | ± | 0.008 | 1.216 | ± | 0.007 | 23.054 | ± | 0.123 | 1.034 | ± | 0.006 | 3.267 | ± | 0.018 |
| $NO_3^-$ | 1.045 | ± | 0.038 | 0.418 | ± | 0.015 | 0.943 | ± | 0.034 | 0.396 | ± | 0.015 | 0.007 | ± | 0.003 | 0.004 | ± | 0.003 | 0.009 | ± | 0.003 |
| $SO_4^{2-}$ | 3.241 | ± | 0.028 | 1.425 | ± | 0.012 | 1.591 | ± | 0.014 | 2.047 | ± | 0.018 | 2.778 | ± | 0.024 | 1.743 | ± | 0.015 | 3.042 | ± | 0.027 |
| **Total (ions)** | 10.410 | | | 4.930 | | | 7.324 | | | 6.831 | | | 42.187 | | | 5.980 | | | 12.496 | | |
| **Total (oxides + ions)** | 58.085 | | | 53.134 | | | 48.658 | | | 74.941 | | | 63.238 | | | 61.527 | | | 66.802 | | |

Note: CaO* and K2O* are water insoluble, $P_2O_5^-$ calculated from total P


## Appendix A

| Sample | DT2_Oct 2015 | DT3_Oct 2015 | DT1_Nov 2015 | DT2_Nov 2015 | DT3_Nov 2015 | DT1_Dec 2015 | DT2_Dec 2015 | DT3_Dec 2015 |
|---|---|---|---|---|---|---|---|---|
| **Major and minor elements as oxides (%)** | | | | | | | | |
| $SiO_2$ | 14.274 ± 0.028 | 38.421 ± 0.068 | 42.440 ± 0.076 | 23.596 ± 0.044 | 38.454 ± 0.070 | 33.248 ± 0.059 | 28.716 ± 0.053 | 28.860 ± 0.052 |
| $TiO_2$ | 0.206 ± 0.001 | 0.587 ± 0.001 | 0.687 ± 0.002 | 0.481 ± 0.001 | 0.651 ± 0.002 | 0.653 ± 0.001 | 0.486 ± 0.001 | 0.507 ± 0.001 |
| $Al_2O_3$ | 3.483 ± 0.048 | 4.611 ± 0.045 | 5.114 ± 0.052 | 3.202 ± 0.046 | 5.353 ± 0.069 | 4.657 ± 0.040 | 3.472 ± 0.055 | 3.603 ± 0.038 |
| $Fe_2O_3$ | 2.118 ± 0.005 | 5.577 ± 0.008 | 6.509 ± 0.010 | 4.736 ± 0.008 | 7.170 ± 0.012 | 6.180 ± 0.009 | 4.276 ± 0.008 | 4.597 ± 0.007 |
| MnO | 0.041 ± 0.002 | 0.103 ± 0.001 | 0.125 ± 0.002 | 0.119 ± 0.002 | 0.121 ± 0.003 | 0.121 ± 0.001 | 0.085 ± 0.002 | 0.079 ± 0.001 |
| CaO* | 5.742 ± 0.008 | 3.357 ± 0.015 | 0.731 ± 0.021 | 1.586 ± 0.010 | 2.370 ± 0.008 | 2.321 ± 0.011 | 1.965 ± 0.015 | 2.502 ± 0.014 |
| $K_2O$* | 0.090 ± 0.010 | 0.774 ± 0.002 | 0.822 ± 0.002 | 0.552 ± 0.002 | 0.954 ± 0.002 | 0.751 ± 0.002 | 0.590 ± 0.003 | 0.592 ± 0.003 |
| $P_2O_5$ | 0.000 ± 0.001 | 0.106 ± 0.001 | 0.000 ± 0.001 | 0.145 ± 0.001 | 0.181 ± 0.002 | 0.133 ± 0.001 | 0.155 ± 0.002 | 0.102 ± 0.001 |
| **Total (oxides)** | 25.953 | 53.537 | 56.429 | 34.418 | 55.254 | 48.064 | 39.745 | 40.840 |
| | | | | | | | | |
| **Trace elements (ppm)** | | | | | | | | |
| V | 67 ± 1 | 125 ± 1 | 130 ± 1 | 149 ± 2 | 180 ± 2 | 144 ± 1 | 97 ± 2 | 65 ± 1 |
| Cr | 26 ± 5 | 110 ± 4 | 119 ± 5 | 300 ± 6 | 135 ± 8 | 218 ± 4 | 103 ± 7 | 71 ± 4 |
| Ni | 33 ± 3 | 84 ± 2 | 68 ± 2 | 971 ± 3 | 121 ± 4 | 211 ± 2 | 134 ± 3 | 64 ± 2 |
| Cu | 28 ± 5 | 111 ± 4 | 59 ± 4 | 102 ± 5 | 74 ± 7 | 78 ± 3 | 89 ± 6 | 55 ± 3 |
| Zn | 82 ± 8 | 250 ± 7 | 205 ± 8 | 507 ± 9 | 358 ± 12 | 534 ± 6 | 328 ± 10 | 231 ± 6 |
| As | 0 ± 5 | 0 ± 4 | 0 ± 5 | 0 ± 5 | 0 ± 8 | 0 ± 3 | 0 ± 7 | 3 ± 4 |
| Br | 457 ± 5 | 27 ± 4 | 56 ± 4 | 66 ± 5 | 61 ± 7 | 59 ± 3 | 112 ± 6 | 95 ± 4 |
| Rb | 11 ± 1 | 40 ± 1 | 35 ± 1 | 26 ± 1 | 34 ± 2 | 27 ± 1 | 22 ± 2 | 22 ± 1 |
| Sr | 227 ± 5 | 422 ± 4 | 683 ± 5 | 253 ± 5 | 210 ± 7 | 329 ± 3 | 266 ± 6 | 310 ± 4 |
| Y | 14 ± 5 | 13 ± 4 | 14 ± 4 | 15 ± 5 | 39 ± 7 | 23 ± 3 | 13 ± 6 | 19 ± 3 |
| Zr | 38 ± 6 | 134 ± 5 | 143 ± 6 | 103 ± 6 | 192 ± 10 | 139 ± 4 | 113 ± 8 | 95 ± 5 |
| Mo | 0 ± 7 | 1 ± 6 | 0 ± 7 | 3 ± 7 | 0 ± 10 | 1 ± 5 | 3 ± 9 | 0 ± 5 |
| Pb | 8 ± 8 | 36 ± 6 | 17 ± 7 | 20 ± 8 | 24 ± 11 | 22 ± 5 | 30 ± 10 | 40 ± 6 |
| | | | | | | | | |
| **Water soluble ions (%)** | | | | | | | | |
| $NH_4^+$ | 0.047 ± 0.005 | 0.043 ± 0.005 | 0.028 ± 0.003 | 0.024 ± 0.003 | 0.025 ± 0.003 | 0.032 ± 0.004 | 0.011 ± 0.001 | 0.025 ± 0.003 |
| $Na^+$ | 10.668 ± 0.073 | 0.577 ± 0.006 | 0.675 ± 0.007 | 0.890 ± 0.008 | 0.684 ± 0.007 | 0.699 ± 0.007 | 1.279 ± 0.010 | 1.540 ± 0.012 |
| $K^+$ | 0.447 ± 0.017 | 0.086 ± 0.003 | 0.106 ± 0.004 | 0.125 ± 0.005 | 0.072 ± 0.003 | 0.081 ± 0.003 | 0.136 ± 0.005 | 0.135 ± 0.005 |
| $Mg^{2+}$ | 1.315 ± 0.018 | 0.164 ± 0.002 | 0.143 ± 0.002 | 0.185 ± 0.003 | 0.066 ± 0.001 | 0.095 ± 0.001 | 0.232 ± 0.003 | 0.225 ± 0.003 |
| $Ca^{2+}$ | 2.543 ± 0.022 | 2.506 ± 0.021 | 1.851 ± 0.016 | 2.059 ± 0.018 | 0.573 ± 0.005 | 1.949 ± 0.017 | 3.422 ± 0.029 | 2.815 ± 0.024 |
| $Cl^-$ | 18.796 ± 0.101 | 0.910 ± 0.005 | 1.181 ± 0.006 | 1.246 ± 0.007 | 1.071 ± 0.006 | 1.120 ± 0.006 | 2.083 ± 0.011 | 2.494 ± 0.013 |
| $NO_3^-$ | 0.009 ± 0.003 | 0.007 ± 0.003 | 0.029 ± 0.003 | 0.004 ± 0.003 | 0.007 ± 0.003 | 0.033 ± 0.003 | 0.205 ± 0.008 | 0.026 ± 0.003 |
| $SO_4^{2-}$ | 3.581 ± 0.031 | 1.411 ± 0.012 | 1.133 ± 0.010 | 2.297 ± 0.020 | 0.697 ± 0.006 | 1.734 ± 0.015 | 2.188 ± 0.019 | 1.964 ± 0.017 |
| **Total (ions)** | 37.407 | 5.704 | 5.146 | 6.831 | 3.196 | 5.745 | 9.558 | 9.224 |
| **Total (oxides + ions)** | 63.360 | 59.240 | 61.574 | 41.249 | 58.450 | 53.810 | 49.303 | 50.065 |

Note: CaO* and $K2O$* are water insoluble, $P_2O_5^-$ calculated from total P