# Peer review of "Physical and chemical properties of deposited airborne particulates over the Arabian Red Sea coastal plain"

_Atmospheric Chemistry and Physics, 2017_

## Referee Comment (RC1) · Anonymous Referee #2 · 8 Jun 2017

**Review on "Physical and chemical properties of deposited airborne particulates over the Arabian Red Sea coastal plain" by Engelbrecht et al., 2017**

The manuscript presents information on monthly resolved dust deposition rates as well as the mineralogical, chemical, and elemental composition of the deposited dust. The data are based on monthly accumulated samples over 13 months from six sites on the campus of King Abdullah University of Science and Technology (KAUST), located on the Saudi Arabian coastal plain near the Red Sea. These are new data from an understudied region, and the provided information is very valuable for other researchers. It will help with a better assessment of the effects of dust from this region on the environment and human health as well as for the evaluation and constraining of dust simulated with models. The manuscript is clearly written and well structured. It should be published after taking into consideration following few minor points.

1. **Page 4, lines 19–25:** Information should be provided where the climatological data were sourced.

2. **Page 5, line 7:** Add *Scanza et al.* (2015) as reference.

3. **Page 8, lines 13–24:** The authors should mention a possible bias in the results from applying the X-ray diffraction (XRD) technique. XRD is most effectively detecting crystalline material. This could lead to an overestimation of the abundance of those dust mineral types that tend to have a regular crystal structure, like tectosilicates, relative to other minerals such as phyllosilicates whose mass can have a significant and varying amorphous fraction (*Formenti et al.*, 2008; *Kandler et al.*, 2009).

4. **Page 8, line 25:** *"Northwesterly Shamal winds prevailed during all twelve months of 2015 (Fig. 3)"*

   What about November? It looks like from Figure 3, as an exception, that northeasterly winds were more frequent in that month, although they didn't quite reach the maximum strength of the northwesterly winds.

**References**

Formenti, P., J. L. Rajot, K. Desboeufs, S. Caquineau, S. Chevaillier, S. Nava, A. Gaudichet, E. Journet, S. Triquet, S. Alfaro, M. Chiari, J. Haywood, H. Coe, , and E. Highwood (2008), Regional variability of the composition of mineral dust from western Africa: Results from the AMMA SOP0/DABEX and DODO field campaigns, *J. Geophys. Res.*, *113*, D00C13, doi:10.1029/2008JD009903.

Kandler, K., L. Schütz, C. Deutscher, M. Ebert, H. Hofmann, S. Jäckel, R. Jaenicke, P. Knippertz, K. Lieke, A. Massling, A. Petzold, A. Schladitz, B. Weinzierl, A. Wiedensohler, S. Zorn, and S. Weinbruch (2009), Size distribution, mass concentration, chemical and mineralogical composition and derived optical parameters of the boundary layer aerosol at Tinfou, Morocco, during SAMUM 2006, *Tellus B*, *61*(1), 32–50, doi:10.1111/j.1600-0889.2008.00385.x.

Scanza, R. A., N. Mahowald, S. Ghan, C. S. Zender, J. F. Kok, X. Liu, Y. Zhang, and S. Albani (2015), Modeling dust as component minerals in the Community Atmosphere Model: development of framework and impact on radiative forcing, *Atmos. Chem. Phys*, *15*, 537–561, doi:10.5194/acp-15-537-2015.

---

## Referee Comment (RC2) · Anonymous Referee #3 · 15 Jun 2017

The paper presents and analyzes the mineralogical, physical and chemical composition of dust deposited samples at six sites on the KAUST campus. KAUST is located on the Red sea coastal plain of Saudi Arabia. Monthly samples were collected between December 2014 and December 2015. The monthly deposition trends were compared to visibility and sun photometric measurements and to previous mineralogical analysis of soil samples from nearby dust sources. The paper concludes that dust deposits along the Red Sea coast are a mixture of dust emissions from local soils and soils imported from distal sources.

As the authors mention in the abstract and the paper, the type of information obtained

has no precedent in the region and can be useful for modelers and other impact areas. This supports publication. In order to make the data useful to others, the authors may include, not only the oxide data in the appendix, but also the mineralogical data and size distribution data. I believe this is the main reason for publication of this manuscript.

While the data will be useful, I find the paper itself overly descriptive. The paper presents the data and a preliminary exploratory data analysis from which it is difficult to extract new insights. The comparison with AERONET, visibility and meteorology is rather superficial. A good example of my argument is Figure 4, where the authors present humidity and temperature data but it is not very clear why they do so. The comparison with AERONET is performed with total AOD. It would have been more appropriate to filter the data by low Angstrom Exponent or use coarse mode AOD.

One of the main conclusions is that dust in the Red Sea is a mixture of dust from local soils and dust imported from distal sources, which is something that is already well-known; the paper makes no attempt to quantify the respective contributions. An additional concern that I have is the selection of the sites to measure dust deposition. It is not very clear what is the impact of local construction activities. The authors should make a clear statement in that respect.

Other than that, I think that the paper is well written and well structured. I didn't find minor errors or typos.

---

## Referee Comment (RC3) · Anonymous Referee #1 · 19 Jun 2017

The present manuscript describes and analyzes the measurement of dust deposition at 6 sites at the King Abdullah University of Science and Technology (KAUST) campus along the Red Sea. The description includes local meteorology and instruments used. The analysis includes size distribution, chemical and mineralogical composition of dust. They compare their results with a previous work performed on soil sample of the same area. It is interesting to see their similarity. They also compare with measurements at other locations in the Arabian Peninsula, Middle East and United States.

These results could quite useful to better characterize dust in the atmosphere. Unfortunately, their use by the modeling community necessitates assumptions, which have

not been discussed. The only thing they provide is a figure showing the number size distribution at one collection site, and they suggest to derive from this figure the mass of particles. This method is inadequate. First, they should provide the values in a Table. Second, this implies assumption concerning shape and density, which varies with soil texture. Third, they should provide variability between sites.

In addition, some work will be necessary to better structure the text, and to clarify some sentences throughout the manuscript. There are also grammatical errors, and typos to correct.

Overall, some efforts have to be done to improve the manuscript and make it more appropriate for publication in Atmospheric Chemistry and Physics, but otherwise it would be a good paper.

Detailed comments: Abstract: Page 2, Line 21-22: "These data will also support dust modeling.. mass balance and optical properties". I wish this would be true. But there is no possibility to derive mass balance from one figure of number size distribution. Concerning optical properties, they are strong function of size distribution. Providing mineralogical data as a function of size will make this paper really useful.

Introduction: The Introduction should be reworked. Some paragraphs in subsequent sections could be moved in the Introduction to improve the reading of the manuscript. I would suggest the following structure, which hopefully help in my following comments. 1. Introduction 1.1 Importance of dust 1.2 Importance of mineralogy 1.3 Previous work on mineralogy 1.4 Gaps 1.5 How is your work filling the gaps 2. Description of the area 2.1. Meteorology 2.2 Dust sources and deposition

Page 3, Line 4. I would rather use Schulz et al. (2012) instead of Bergametti and Foret (2014). It is a more appropriate citation for uncertainties associated with model dust deposition. Page 3. Line 9: "..important dust source regions". You may want to cite the comprehensive work on the subject by Prospero et al. (2002) and Ginoux et al. (2012) Page 3 Line17-21: Limit the number of citations to key papers. Page 4, Line

19 –Page 5, Line 4: this paragraph does not fit in the flow of thinking. I suggest to move it in the proposed Section 2.1 providing description of the general area. Page 4, Line 5: You never say why mineralogy is important, although this should be the key motivation of this work. You should develop this into a full paragraph (proposed Section 1.2). Page 5, Line 9: "However" remove Page 5, Line 12. Break the sentence after the citations, and replace "varying with" by "Its adverse effects will depend on" Page 5 Objectives: This should be articulate within the Introduction. Start by saying why mineralogy is important, then what has been done, then what is the originality of the work, and then finish by providing a succinct outline of the manuscript. Page 5, Line 25: "plain to be an" => "plain is an" Page 5, Line 25: remove "province" Page 5, Line 25-26: sentence unclear, and provide a reference. Page 5, Line 27: remove "inevitably" Page 6, Line 8-18: you repeat yourself. Restructure as suggested. Page 6: Line 20-25: Move to suggested Section 2.2 where you describe the general area. Page 8, Line 15 "soils and dusts" replace (as well as all other occurrences) by "soil and dust". Page 8, Line 16-18. This is an argument showing the importance of mineralogy and should be moved in suggested Section 1.2 Page 8, Line 19-23: It is unclear what are these 3 methods for. Are they all used for mineralogical analysis? What are the benefits of using 3 methods? Page 9, Line 1 to 14. I don't see the relevance between your measurement and ambient temperatures. Does it matter? On the other hand, did it rain anytime? Page 9, "Gravimetric Analysis": In Figure 5, you did not discuss the peak in dust deposition in August in DT3. This maximum is 3 times higher than the annual mean, and 30% higher than DT4. Why such difference between DT3 and DT4 and all other sites in August? This factor 3 difference will affect your analysis, but first you will have to know its origin. Is it construction? What is the mineralogy or chemical components of construction dust? Do you see its signature in your data? Page 10, Line 5-7. Reformulate the sentence. => The dust deposition measured in Kuwait on the other hand, varies substantially between sites due to the contribution from disturbed soils in lowlands during periods of northwesterly Shamal. Page 10, Line 8-13: Remove, this is repeating what is already in Table 3. Page 10, AERONET:

[Figure]

You should either use the Ansgtrom exponent to screen out non-dusty days or use SDA coarse mode optical depth. Page 10, Line 26: "dust particles are predominantly from local sources" but in Abstract you wrote "dust deposits along the Red Sea coast are a mixture of dust emission from local soils, and soils imported from distal sources." This is contradictory. Page 10 Line 28-30: You should remove and screen AERONET data using low Angstrom values, or use AERONET SDA coarse mode AOD. Page 11, Line 10: Merge Figure 6 and 7. Page 11, Line 13-15: This is irrelevant for this study. Remove. Page 11, Mineral analysis: A point that needs clarification is the units. Are all the % values given by mass or by number? In section 4.6, it is specified by mass. This means that you should be able to provide the mass size distribution! Page 12, Line 11: "DT1". Why only one site and not all of them? Why is there no standard deviation in Figure 9a. What is the error associated with these measurements? Page 12, Line 21-22. "... Figure 9 could be used to distinguish the contribution of PM10 in deposited mass and reconcile models with observations." Are you suggesting that modelers use a ruler to derive approximately some fraction of particle numbers, then assume some density and shape for each sizes? This is an inadequate method. You should provide the values of each dots of Figure 9a in a Table, as well as the errors associated with the measurement, and assumptions on shape and density. Page 13, Line 3 "soils and dusts" => "soil and dust" Page 14, Line 13: "This paper has as its goal the provision" Needs to be reformulated Page 14, Line 16: "..meant to be used for validating dust mass balance.." No. The method suggested in Section 4.5 is inadequate. Page 15, Line 3-4 contradicts Line 5-6. Page 15, Line 12: you may want to add "construction dust". Page 15, Line 13: "To better represent.." In what sense? By models? This may be a good place to add that the "inclusion of particle size into mineralogical and chemical analysis will provide more effectively data for the modeling community." Page 25, Line 4: "Locality" => Position Page 25, Line 5: "campus...Sea" => on the Arabian Peninsula (red dot) Page 27: Provide a Figure caption rather than an analysis of the Figure. Page 28: add the color of each lines in the Figure caption in parenthesis. Page 30 & 31: Merge the 2 Figures. Page 32: Is there a possibility to split between fine,

coarse and super-coarse modes? Page 33, Figure 9a: Error bars

References: Ginoux, P., Prospero, J.M., Gill, T.E., Hsu, N.C. and Zhao, M., 2012. Global‐scale attribution of anthropogenic and natural dust sources and their emission rates based on MODIS Deep Blue aerosol products. Reviews of Geophysics, 50(3). Prospero, J.M., Ginoux, P., Torres, O., Nicholson, S.E. and Gill, T.E., 2002. Environmental characterization of global sources of atmospheric soil dust identified with the Nimbus 7 Total Ozone Mapping Spectrometer (TOMS) absorbing aerosol product. Reviews of geophysics, 40(1). Schulz, M., Prospero, J.M., Baker, A.R., Dentener, F., Ickes, L., Liss, P.S., Mahowald, N.M., Nickovic, S., García-Pando, C.P., Rodríguez, S. and Sarin, M., 2012. Atmospheric transport and deposition of mineral dust to the ocean: implications for research needs. Environmental science & technology, 46(19), pp.10390-10404.

———————————————————

---

## Author Comment (AC2) · 1 Aug 2017

Atmos. Chem. Phys. Discuss. https://doi.org/10.5194/acp-2017-231-RC3, 2017 © Author(s) 2017. This work is distributed under the Creative Commons Attribution 3.0 License Interactive comment on "Physical and chemical properties of deposited airborne particulates over the Arabian Red Sea coastal plain" by Johann Engelbrecht et al. Anonymous Referee #2 The manuscript presents information on monthly resolved dust deposition rates as well as the mineralogical, chemical, and elemental composition of the deposited dust. The data are based on monthly accumulated samples over 13 months from six sites on the campus of King Abdullah University of Science and

[Figure]

Technology (KAUST), located on the Saudi Arabian coastal plain near the Red Sea. These are new data from an understudied region, and the provided information is very valuable for other researchers. It will help with a better assessment of the effects of dust from this region on the environment and human health as well as for the evaluation and constraining of dust simulated with models. The manuscript is clearly written and well structured. It should be published after taking into consideration following few minor points.

1. Page 4, lines 19-25: Information should be provided where the climatological data were sourced.

Authors Response: Data sources added.

With the exception of the area around Jazan in the south, which is impacted by the Indian Ocean monsoon, the Red Sea coastal region has a desert climate characterized by extreme heat. Temperatures measured at the KAUST campus reach 43° C during the summer days, with a drop in night-time temperatures on average of more than 10° C. Also, although the extreme temperatures here are moderated by the proximity of the Red Sea, summer humidity is often 85 % or higher during periods of the northwesterly Shamal winds. Rainfall diminishes from an annual average of 133 mm at Jazan in the south to 56 mm at Jeddah, and 30 mm at Tabuk in the north http://worldweather.wmo.int/en/city.html?cityId=699.

2. Page 5, line 7: Add Scanza et al. (2015) as reference.

Authors Response: Reference was added.

3. Page 8, lines 13-24: The authors should mention a possible bias in the results from applying the X-ray diffraction (XRD) technique. XRD is most effectively detecting crystalline material. This could lead to an overestimation of the abundance of those dust mineral types that tend to have a regular crystal structure, like tectosilicates, relative to other minerals such as phyllosilicates whose mass can have a significant and varying

amorphous fraction (Formenti et al., 2008; Kandler et al., 2009).

Authors Response: Added text following line 24.

A likely bias in the results from applying the X-ray diffraction (XRD) technique together with the RIR method is widely recognized, and therefore our methodology is considered to be semi-quantitative at best. Chung (1974) recognized that if the RIRs of all the crystalline phases in a mineral mixture are known, the sum of all the fractions should add to 100%. However, XRD is effective at measuring crystalline phases such as quartz, calcite, and feldspars, and less so for partly crystalline and amorphous phases including some layered silicates such as clays as well as many hydrous minerals. This could lead to an overestimation of the abundance of the crystalline mineral types in the dust, compared to partly crystalline and amorphous phases (Formenti et al., 2008; Kandler et al., 2009). Other discrepancies could occur from preferred orientation of layered silicates in the sample mounts, and the dust samples were loaded into side-mount holders to minimize this effect.

4. Page 8, line 25: "Northwesterly Shamal winds prevailed during all twelve months of 2015 (Fig. 3)" What about November? It looks like from Figure 3, as an exception, that northeasterly winds were more frequent in that month, although they didn't quite reach the maximum strength of the northwesterly winds.

Authors'Response: Sentence added.

Although the northeasterly winds were more frequent in November, they did not reach the strength of the northwesterlies.

References Formenti, P., J. L. Rajot, K. Desboeufs, S. Caquineau, S. Chevaillier, S. Nava, A. Gaudichet, E. Journet, S. Triquet, S. Alfaro, M. Chiari, J. Haywood, H. Coe, and E. Highwood (2008), Regional variability of the composition of mineral dust from western Africa: Results from the AMMA SOP0/DABEX and DODO field campaigns, J. Geophys. Res., 113, D00C13, doi:10.1029/2008JD009903.

Kandler, K., L. Schütz, C. Deutscher, M. Ebert, H. Hofmann, S. Jäackel, R. Jaenicke, P. Knippertz, K. Lieke, A. Massling, A. Petzold, A. Schladitz, B. Weinzierl, A. Wiedensohler, S. Zorn, and S. Weinbruch (2009), Size distribution, mass concentration, chemical and mineralogical composition and derived optical parameters of the boundary layer aerosol at Tinfou, Morocco, during SAMUM 2006, Tellus B, 61 (1), 32{50, doi:10.1111/j.1600-0889.2008.00385.x.

Scanza, R. A., N. Mahowald, S. Ghan, C. S. Zender, J. F. Kok, X. Liu, Y. Zhang, and S. Albani (2015), Modeling dust as component minerals in the Community Atmosphere Model: development of framework and impact on radiative forcing, Atmos. Chem. Phys., 15, 537-561, doi:10.5194/acp-15-537-2015.

---

## Author Comment (AC3) · 1 Aug 2017

The paper presents and analyzes the mineralogical, physical and chemical composition of dust deposited samples at six sites on the KAUST campus. KAUST is located on the Red sea coastal plain of Saudi Arabia. Monthly samples were collected between December 2014 and December 2015. The monthly deposition trends were compared to visibility and sun photometric measurements and to previous mineralogical analysis of soil samples from nearby dust sources. The paper concludes that dust deposits along the Red Sea coast are a mixture of dust emissions from local soils and soils imported from distal sources.

As the authors mention in the abstract and the paper, the type of information obtained has no precedent in the region and can be useful for modelers and other impact areas. This supports publication.

1. In order to make the data useful to others, the authors may include, not only the oxide data in the appendix, but also the mineralogical data and size distribution data. I believe this is the main reason for publication of this manuscript.

Authors Response: Particle size distribution plots of 12 deposition samples collected monthly at the KAUST campus throughout the 2015 period are added as Appendix A. The mineralogical data are added as a table in Appendix B. The chemical data tables are renumbered as Appendix C.

2. While the data will be useful, I find the paper itself overly descriptive. The paper presents the data and a preliminary exploratory data analysis from which it is difficult to extract new insights. The comparison with AERONET, visibility and meteorology is rather superficial. A good example of my argument is Figure 4, where the authors present humidity and temperature data but it is not very clear why they do so. The comparison with AERONET is performed with total AOD. It would have been more appropriate to filter the data by low Angstrom Exponent or use coarse mode AOD.

Authors Response: : This is discussed under items 27 and 29 of authors response to referee #1 comments.

[Figure]

3. One of the main conclusions is that dust in the Red Sea is a mixture of dust from local soils and dust imported from distal sources, which is something that is already well-known; the paper makes no attempt to quantify the respective contributions. An additional concern that I have is the selection of the sites to measure dust deposition. It is not very clear what is the impact of local construction activities. The authors should make a clear statement in that respect.

Authors Response: Source apportionment is considered to be a further step in our research, to be documented in a following paper. As an approximation the sampler with the lowest deposition rate can be considered to have negligible or the least amount of local dust and sea salt. In the months of December 2014, January, April, March, June, July, and December 2015, the deposition rates at the four sites were similar, and considered to have no or negligible amounts of dust from local construction, campus roads, marine salt, or other particulates.

Other than that, I think that the paper is well written and well structured. I did not find minor errors or typos.

---

## Author Comment (AC4) · 1 Aug 2017

Atmos. Chem. Phys. Discuss. https://doi.org/10.5194/acp-2017-231-RC3, 2017 ©
The present manuscript describes and analyzes the measurement of dust deposition at 6 sites at the King Abdullah University of Science and Technology (KAUST) campus along the Red Sea. The description includes local meteorology and instruments used. The analysis includes size distribution, chemical and mineralogical composition of dust. They compare their results with a previous work performed on soil sample of the same area. It is interesting to see their similarity. They also compare with measurements at other locations in the Arabian Peninsula, Middle East and United States. 1. These results could quite useful to better characterize dust in the atmosphere. Unfortunately, their use by the modeling community necessitates assumptions, which have not been discussed. The only thing they provide is a figure showing the number size distribution at one collection site, and they suggest to derive from this figure the mass of particles. This method is inadequate. First, they should provide the values in a Table. Second, this implies assumption concerning shape and density, which varies with soil texture. Third, they should provide variability between sites. Authors Response: We have made adjustments as suggested below. 2. In addition, some work will be necessary to better structure the text, and to clarify some sentences throughout the manuscript. There are also grammatical errors, and typos to correct. Overall, some efforts have to be done to improve the manuscript and make it more appropriate for publication in Atmospheric Chemistry and Physics, but otherwise it would be a good paper. Authors' Response: We have made adjustments as suggested below. 3. Detailed comments: Abstract: Page 2, Line 21-22: "These data will also support dust modeling.. mass balance and optical properties". I wish this would be true. But there is no possibility to derive mass balance from one figure of number size distribution. Concerning optical properties, they are strong function of size distribution.

Authors Response: We replaced the single size distribution plot (Fig. 9a) with comparable 12 monthly size distribution plots, expressed both as number (Supplement C) and volume percentages (Figure 8a, Appendix A). A table (Table 4) with distribution statistics, and assessments of the $<10\mu$m and $<2.5\mu$m mass fractions are given. The table also provides information on particle deposition rates by month. 4. Providing

mineralogical data as a function of size will make this paper really useful. Authors Response: Mineralogical data as measured by XRD per size fraction is not available, only for the total deposition samples (Fig. 7) 5. Introduction: The Introduction should be reworked. Some paragraphs in subsequent sections could be moved in the Introduction to improve the reading of the manuscript. I would suggest the following structure, which hopefully help in my following comments. 1. Introduction 1.1 Importance of dust 1.2 Importance of mineralogy 1.3 Previous work on mineralogy 1.4 Gaps 1.5 How is your work filling the gaps 2. Description of the area 2.1. Meteorology 2.2 Dust sources and deposition. Authors Response: The chapters 1 and 2 were reworked as suggested to improve the reading of the manuscript 6. Page 3, Line 4. I would rather use Schulz et al. (2012) instead of Bergametti and Foret (2014). It is a more appropriate citation for uncertainties associated with model dust deposition. Authors Response: We added Schultz et al (2012) to the Bergametti and Foret (2014) reference 7. Page 3. Line 9: "important dust source regions". You may want to cite the comprehensive work on the subject by Prospero et al. (2002) and Ginoux et al. (2012) Authors Response: Added references Prospero et al. (2002) and Ginoux et al. (2012) 8. Page 3 Line17-21: Limit the number of citations to key papers. Page 4, Line 19 –Page 5, Line 4: this paragraph does not fit in the flow of thinking. I suggest to move it in the proposed Section 2.1 providing description of the general area. Authors Response: The references will be sorted under each category but retained as such. Moved to new Section 2.1 9. Page 4, Line 5: You never say why mineralogy is important, although this should be the key motivation of this work. You should develop this into a full paragraph (proposed Section 1.2). Authors Response: Expanded in new Section 1.2. Mineralogy is important in the dust forming process in soils, and the dust transport mechanism. Optical properties such as refractive indices differ amongst minerals. 10. Page 5, Line 9: "However" remove Authors Response: "However" removed 11. Page 5, Line 12. Break the sentence after the citations, and replace "varying with" by "Its adverse effects will depend on" Authors Response: Sentence restructured as suggested 12. Page 5 Objectives: This should be articulate within the Introduction. Start by saying why mineralogy is

important, then what has been done, then what is the originality of the work, and then finish by providing a succinct outline of the manuscript. Authors Response: Objectives included in Introduction 13. Page 5, Line 25: "plain to be an" => "plain is an" Authors Response: Corrected as suggested 14. Page 5, Line 25: remove "province" Authors Response: "province" removed 15. Page 5, Line 25-26: sentence unclear, and provide a reference. Authors Response: Rephrased and references, Prakash et al, Anatolii et al provided 16. Page 5, Line 27: remove "inevitably" Authors Response: "inevitably" removed 17. Page 6, Line 8-18: you repeat yourself. Restructure as suggested. Authors Response: Restructured? 18. Page 6: Line 20-25: Move to suggested Section 2.2 where you describe the general area. Authors Response: Moved to suggested Section 2.2 19. Page 8, Line 15 "soils and dusts" replace (as well as all other occurrences) by "soil and dust". Authors Response: Replace with "soil and dust" although the plural "soils and dusts" is not incorrect 20. Page 8, Line 16-18. This is an argument showing the importance of mineralogy and should be moved in suggested Section 1.2 Authors Response: Lines 16-18 moved to the new Section 1.2 21. Page 8, Line 19-23: It is unclear what are these 3 methods for. Are they all used for mineralogical analysis? What are the benefits of using 3 methods? Authors Response: This comment is unclear. The three methods measure different components of the dust, XRF for chemical elements, IC for the ions, and XRD for the mineral phases. 22. Page 9, Line 1 to 14. I don't see the relevance between your measurement and ambient temperatures. Does it matter? On the other hand, did it rain anytime? Authors Response: In the light of the current study we find it relevant to provide a brief description of weather conditions at the measurement sites. High ambient temperatures is an indicator of highly turbulent conditions, favorable for dust transport and increased deposition. We do not discuss wet deposition of dust directly, but we note that hazy humid conditions during some mornings probably contribute to the deposition of dust, justifying the discussion of the dewpoint. In 2015, there were only a few light rainfall events at KAUST, and as such not of much importance to our measurements. 23. Page 9, "Gravimetric Analysis": In Figure 5, you did not discuss the peak in dust deposition in August in DT3. This

maximum is 3 times higher than the annual mean, and 30% higher than DT4. Why such difference between DT3 and DT4 and all other sites in August? This factor 3 difference will affect your analysis, but first you will have to know its origin. Is it construction? Authors Response: Added sentence. The higher deposition rate of DT3 for August compared to DT4 is ascribed to the fact that the former is about 100m closer to construction material handling activities during that month. 24. What is the mineralogy or chemical components of construction dust? Do you see its signature in your data? Authors Response: We do not have chemical or mineralogical signatures for construction dust. However, it is expected to contain variable amounts of local dirt road dust, sand, and cement products. The composition thereof will vary substantially with each construction activity. The best we can do is to ascribe elevated mass concentration at any one site to the contribution by construction dust (or marine salt in some cases. This explains the anomalously high concentrations at individual sites (Figure 5, August, September, October). 25. Page 10, Line 5-7. Reformulate the sentence. => The dust deposition measured in Kuwait on the other hand, varies substantially between sites due to the contribution from disturbed soils in lowlands during periods of northwesterly Shamal. Authors Response: Rephrased to read "The dust deposition measured in Kuwait on the other hand, varies substantially between sites due to the contribution from disturbed soils in lowlands during periods of northwesterly Shamal winds." 26. Page 10, Line 8-13: Remove, this is repeating what is already in Table 3. Authors Response: Removed these lines 27. Page 10, AERONET: You should either use the Angstrom exponent to screen out non-dusty days or use SDA coarse mode optical depth. Authors Response: The KAUST campus is located in the heart of the dust source region. The average optical depth is about 0.4 and there are virtually no non-dusty days. According to CALIPSO, the ratio of "not dust" to "dust" successful retrievals in this region is 2.04% indicating that dust dominates all other types of aerosol (Osipov et al., 2015). Kalenderski and Stenchikov (2016) demonstrated that over the Arabian Peninsula the contribution of non-dust aerosols in the visible optical depth does not exceed 10%. Khan et al. (2015) compared the contributions of fine and

coarse modes of aerosols over the similar dust source region in Sahara and showed that the coarse mode dominates the optical depth. Having said this, we consider it unnecessary to further recalculate the AERONET observations to improve the correlation between deposition and AOD because, as we mentioned in the paper, there are other important factors that could affect this relation, which in the context of this paper is treated only qualitatively 28. Page 10, Line 26: "dust particles are predominantly from local sources" but in Abstract you wrote "dust deposits along the Red Sea coast are a mixture of dust emission from local soils, and soils imported from distal sources." This is contradictory. Authors Response: Replaced "predominantly" with "partly" 29. Page 10 Line 28-30: You should remove and screen AERONET data using low Angstrom values, or use AERONET SDA coarse mode AOD. Authors Response: See response under comment 27 30. Page 11, Line 10: Merge Figure 6 and 7. Authors Response: Since the three variables (deposition rate, visibility, AOD) have different units, it will be confusing to the reader have them together on a single plot. The two diagrams (Figures 6, 7) are combined as Figure 6a and 6b. 31. Page 11, Line 13-15: This is irrelevant for this study. Remove. Authors Response: Sentence and references removed 32. Page 11, Mineral analysis: A point that needs clarification is the units. Are all the % values given by mass or by number? In section 4.6, it is specified by mass. This means that you should be able to provide the mass size distribution! Authors Response: Correct. The mineral analysis by XRD and chemistry are all mass percentages. To be comparable, the particle size distributions are now presented and discussed as both mass (volume) (Appendix A) and number (Supplement C). 33. Page 12, Line 11: "DT1". Why only one site and not all of them? Why is there no standard deviation in Figure 9a. What is the error associated with these measurements? Authors Response: See below. Additional SEM analysis were recently performed and are now included (Appendix A, Supplement C, Table 4) 34. Page 12, Line 21-22. ". . . Figure 9 could be used to distinguish the contribution of PM10 in deposited mass and reconcile models with observations." Are you suggesting that modelers use a ruler to derive approximately some fraction of particle numbers, then assume some density and shape for

each sizes? This is an inadequate method. You should provide the values of each dots of Figure 9a in a Table, as well as the errors associated with the measurement, and assumptions on shape and density. Authors Response: We replaced the single size distribution plot (Fig. 9) with comparable 12 monthly size distribution plots, expressed both as number and volume percentages (Appendix A, Supplement C, Table 4). It was shown that the deposition rates at all the sites are similar, except for the added contributions from local marine and local construction at some sites. A table (Table 4) with distribution statistics, assessments of the <2.5 $\mu$m and <10$\mu$m mass fractions are presented. The table also provides information on particle size distribution variability by month.

35. Page 13, Line 3 "soils and dusts" => "soil and dust" Authors Response: Corrected to read "soil and dust" 36. Page 14, Line 13: "This paper has as its goal the provision" Needs to be reformulated Authors Response: Rephrased to read "This paper provides new mineralogical, physical and chemical results on deposition samples collected at the KAUST campus during 2015". 37. Page 14, Line 16: "meant to be used for validating dust mass balance.." No. The method suggested in Section 4.5 is inadequate. Authors Response: Deleted "and is meant to be used for validating dust mass balance in the meteorological models with the dust component". 38. Page 15, Line 3-4 contradicts Line 5-6. Authors Response: Deleted the sentence in line 3 and 4 reading "It is therefore not feasible to explicitly relate the deposition samples to the coastal soils from chemical and mineralogical results on their own." 39. Page 15, Line 12: you may want to add "construction dust". Authors Response: Added " ,local construction dust.." 40. Page 15, Line 13: "To better represent.." In what sense? By models? This may be a good place to add that the "inclusion of particle size into mineralogical and chemical analysis will provide more effectively data for the modeling community." Authors Response: Replaced "represent" with "To better model the dust being deposited . . . . . . .". Added as suggested "Also, inclusion of particle size with mineralogical and chemical measurements will provide more effective data for the modeling community." 41. Page 25, Line 4: "Locality" => Position Authors Response: Rephrased as suggested 42. Page 25, Line 5: "campus. . .Sea" => on the Arabian Peninsula (red dot) Authors Response: Rephrased as suggested 43. Page 27: Provide a Figure caption rather than an analysis of the Figure. Authors Response: Unsure what is meant with analysis of the Figure 3. 44. Page 28: add the color of each lines in the Figure caption in parenthesis. Authors Response: Parenthesis added in Figure 4 caption 45. Page 30 & 31: Merge the 2 Figures. Authors Response: Since the three variables (deposition rate, visibility, AOD) have different units, it will be confusing to the reader have them together on a single plot. The two diagrams (Figures 6,7) are retained as such, but combined as one figure (Fig. 6a,b). 46. Page 32: Is there a possibility to split between fine, coarse and super-coarse modes? Authors Response: The 0.2 -2.5 $\mu$m, 0.2-10 $\mu$m and >10$\mu$m mass percentages were assessed from the mass percentage/particle size distribution plots, shown in Table 4. 47. Page 33, Figure 9a: Error bars. Authors Response: Figure 9a is replaced by an averaged plot for the 12-month sampling period, including the uncertainty bars (Fig 8a).

References: Ginoux, P., Prospero, J.M., Gill, T.E., Hsu, N.C. and Zhao, M., 2012. Global-scale attribution of anthropogenic and natural dust sources and their emission rates based on MODIS Deep Blue aerosol products. Reviews of Geophysics, 50(3). Prospero, J.M., Ginoux, P., Torres, O., Nicholson, S.E. and Gill, T.E., 2002. Environmental characterization of global sources of atmospheric soil dust identified with the Nimbus 7 Total Ozone Mapping Spectrometer (TOMS) absorbing aerosol product. Reviews of geophysics, 40(1). Schulz, M., Prospero, J.M., Baker, A.R., Dentener, F., Ickes, L., Liss, P.S., Mahowald, N.M., Nickovic, S., García-Pando, C.P., Rodríguez, S. and Sarin, M., 2012. Atmospheric transport and deposition of mineral dust to the ocean: implications for research needs. Environmental science & technology, 46(19), pp.10390-10404. Kalenderski, S., and Stenchikov, G.: High-resolution regional modeling of summertime transport and impact of African dust over the Red Sea and Arabian Peninsula, Journal of Geophysical Research: Atmospheres, 121, 6435–6458, doi:10.1002/2015JD024480, 2016. Khan, B., Stenchikov, G., Weinzierl, B., Kalenderski, S., and Osipov, S.: Dust plume formation in the free troposphere and aerosol

size distribution during the Saharan Mineral Dust Experiment in North Africa, Tellus B: Chemical and Physical Meteorology, 67, 1, 27170, doi: 10.3402/tellusb.v67.27170, 2015. Osipov, S., Stenchikov, G., Brindley, H., and Banks, J.: Diurnal cycle of the dust instantaneous direct radiative forcing over the Arabian Peninsula, Atmospheric Chemistry and Physics, 15, 9537–9553, doi:10.5194/acp-15-9537-2015, 2015.

---

## Author Comment (AC5) · 1 Aug 2017

10    **Anonymous Referee #1**

[revised manuscript text omitted]